# Inhibition of HDAC6 alters fumarate hydratase activity and mitochondrial structure

Andrew Roe[1], Catríona M. Dowling[2], Cian D'Arcy[1], Daniel Alencar Rodrigues [1], Yu Wang[1], Matthew Hiller [1], Carl Keogh [1], Kate E. R. Hollinshead[3], Massimiliano Garre [4], Brenton Cavanagh [5], Kieran Wynne[6,7], Tianyan Liu[8], Zhixing Chen [8], Emma Kerr [9], Marie McIlroy[10], Jochen H. M. Prehn [1], Ingmar Schoen [11] & Tríona Ní Chonghaile [1] ✉

Fumarate hydratase (FH), a key node of mitochondrial metabolism, is also a tumour suppressor. Despite its prominent roles in tumourigenesis and inflammation, its regulation remains poorly understood. Herein, we show that histone deacetylase 6 (HDAC6) regulates FH activity. In triple-negative breast cancer cells, HDAC6 inhibition or knockdown results in alterations to mitochondrial cristae structure, as detected by live-cell super-resolution STED nanoscopy and electron microscopy, along with the release of mitochondrial DNA. Mass-spectrometry immunoprecipitation reveals multiple mitochondrial HDAC6-interactors, with FH emerging as a top hit. Super-resolution 3D-STORM shows HDAC6 interactions with FH in mitochondrial networks, which increases after perturbation of HDAC6 activity with BAS-2. Treatment with BAS-2 leads to fumarate accumulation by $^{13}C$ glucose labelling, along with downstream succination of proteins and cell death. Together, these results identify HDAC6 inhibition as a regulator of endogenous FH activity in tumour cells, and highlight it as a promising candidate for indirectly targeting tumour metabolism.

Altered metabolism is a key hallmark of tumour cells[1] and has long been associated with relapse and resistance to treatment across different tumour types[2,3]. Histone deacetylase 6 (HDAC6) was thought of as a cytosolic HDAC that regulated microtubule dynamics through the deacetylation of α-tubulin and cortactin[4,5]. The only HDACs previously associated with metabolism were the sirtuins (SIRT3-5), which regulated metabolic functions in the mitochondria through NAD$^+$-dependent post-translational modifications[6]. Recently, our lab identified that

HDAC6 deacetylated numerous metabolic proteins and that HDAC6 inhibition with the newly identified inhibitor BAS-2 or HDAC6 knockdown altered glycolytic metabolism in triple-negative breast and lung cancer[7]. Interestingly, we also found that HDAC6 interacted with a substantial amount of mitochondrial proteins, including fumarate hydratase (FH).

FH is a key enzyme in the mitochondrial tricarboxylic acid (TCA) cycle that catalyses the conversion of fumarate into L-malate[8,9]. FH is a

[1]Department of Physiology and Medical Physics, Royal College of Surgeons in Ireland, Dublin, Ireland. [2]Department of Medicine, Royal College of Surgeons in Ireland, Dublin, Ireland. [3]Department of Radiation Oncology, Laura and Isaac Perlmutter Cancer Center, New York University Langone Medical Center, New York, NY, USA. [4]Super Resolution Imaging Consortium (SRIC), Department of Chemistry, Royal College of Surgeons in Ireland, Dublin, Ireland. [5]Cellular & Molecular Imaging Core, Royal College of Surgeons in Ireland, Dublin, Ireland. [6]Systems Biology Ireland, School of Medicine, University College Dublin, Belfield, Dublin, Ireland. [7]UCD Conway Institute, University College Dublin, Dublin, Ireland. [8]College of Future Technology, Peking University, Beijing, China. [9]Patrick G Johnson Centre for Cancer Research, Queen's University, Belfast, N, Ireland. [10]Department of Surgery, Royal College of Surgeons in Ireland, Dublin, Ireland. [11]School of Pharmacy and Biomolecular Sciences, Royal College of Surgeons in Ireland, Dublin, Ireland. ✉e-mail: tnichonghaile@rcsi.ie

nuclear-encoded protein, with two transcripts, one of which has an amino signal sequence that directs it to the mitochondria, where it plays a major role in the TCA cycle. The mitochondrial TCA cycle is a series of reactions in a closed circuit that drives metabolism within cells[10]. The TCA cycle plays an important role in tumour biology, not only for energy production through the respiratory chain, but also for metabolic intermediates that serve as building blocks for lipid and nucleic acid synthesis, enabling proliferation[11].

Some of these intermediates of the TCA cycle also have dual roles as 'oncometabolites'[12]. Both fumarate hydratase (FH) and succinate dehydrogenase (SDH), two key enzymes in the TCA cycle, are known tumour suppressors[8,13]. Loss of *FH* through germline inactivating mutations is associated with an aggressive form of kidney cancer called hereditary leiomyomatosis and renal cell carcinoma (HLRCC)[14]. Deficiency of FH disrupts the TCA cycle, leading to fumarate accumulation, which has various signalling properties, including the non-enzymatic formation of 2-(succino)cysteine (2-SC) adducts that can alter protein activity, a process termed succination[15–17]. Two recent publications identified that macrophages and kidney epithelial cells with FH deletion show an increase of intracellular fumarate and that this alters mitochondrial structure, causing the release of mitochondrial DNA or RNA to activate inflammatory responses[18,19]. To date, the majority of studies on FH have been carried out using knockout mice to show the role of FH in tumourigenesis or macrophage inflammation[19,20]. However, there are limited studies on the role of FH in tumour cells or on how FH activity is regulated. It remains unclear how cells with altered FH activity can rewire their metabolism to complete the TCA cycle and whether an induced therapeutic vulnerability can be exposed.

In this study, we identified that HDAC6 can cause alterations to mitochondrial cristae structure, causing cristae restructuring and the release of mtDNA into the cytosol. We then show by mass spectrometry, immunoprecipitation, and super-resolution imaging that HDAC6 and FH directly interact. Inhibition of HDAC6 with BAS-2 reduces FH activity, resulting in an increase of fumarate and downstream protein succination. Inhibition of FH activity induces a reliance on pyruvate decarboxylase, facilitating an alternative pathway for malate production. Utilising stochastic optical reconstruction microscopy (STORM), we pinpoint the sites of interaction to the mitochondria, thereby demonstrating a regulation of FH activity in tumour cells following HDAC6 inhibition.

## Results

### HDAC6 inhibition or knockdown causes altered mitochondrial structure in TNBC

Following the discovery of the HDAC6 inhibitor, BAS-2, and the identification of the role HDAC6 played in glycolytic metabolism[21], we first sought to measure if HDAC6 activity altered mitochondrial structure or phenotype. We evaluated this using transmission electron microscopy (TEM) in a TNBC cell line where BAS-2 was originally discovered to be cytotoxic[21]. Treatment with the HDAC6 inhibitor BAS-2 caused notable changes in mitochondrial ultrastructure of MDA-MB-231 cells (Fig. 1A), at a dose that results in HDAC6 inhibition with no significant effects on cell death (10 μM, Supplementary Fig. 1A–F). Mitochondrial cross-sectional area increased (Fig. 1B), indicating mitochondrial fusion or swelling. Most notably, cristae appeared disrupted with treatment, indicating inner membrane damage or increased fragility (Fig. 1C). Change in mitochondrial size was confirmed by MitoTracker Green staining for mitochondrial mass, although it was only significant at the higher-dose condition (Fig. 1D). To confirm that the effects were HDAC6-mediated, we generated CRISPR/Cas9-mediated HDAC6 knockdown MDA-MB-231 cells (sgHDAC6A, B). Validating the HDAC6 K/D in cells, increased lysine 40 (K40) acetylation of α-tubulin, a substrate of HDAC6, was detected by Western blot (Fig. 1E). In comparison to the control cells, HDAC6 K/D cells also exhibited increased

mitochondrial area and cristae disruption (Fig. 1F–H), indicating mitochondrial effects as an HDAC6-mediated phenotype.

### HDAC6 inhibition induced mtDNA release and increased mtROS in TNBC cells

As cells are required to be chemically or cryogenically fixated for electron microscopy, we next assessed mitochondrial disruption in living cells using stimulated emission depletion (STED) super-resolution microscopy in a second TNBC cell line. The BT-549 cell line was chosen due to the lower motility and larger mitochondria as compared to MDA-MB-231, which makes them more suitable for live-cell imaging of cristae. Mitochondria stained with PKMitoOrange[22] (PKMO) are shown in orange using STED imaging and mtDNA is stained with picogreen, shown in cyan, using standard confocal mode (Fig. 2A). As was found in the TEM images, there was evidence of mitochondrial cristae disruption following low-dose treatment with BAS-2. In addition, there seems to be an expansion of the stain for mtDNA and the appearance of cytosolic DNA (Fig. 2B). This was quantified by measuring picogreen pixel count outside of the nucleus, which includes mitochondrial nucleoids and cytosolic DNA. There was a dose-dependent increase in picogreen pixel count following BAS-2 treatment (Fig. 2C). At concentrations of BAS-2 that can cause cell death (Supplementary Fig. 1A, D), there is evidence of hyperfused mitochondria along with an increase in mtDNA in the cytosol (Fig. 2A–C). To confirm that cytosolic DNA came from the mitochondria and not the nucleus, a co-stain of mitochondrial transcription factor A (TFAM), was carried out by immunofluorescence. TFAM is the most abundant component of mitochondrial nucleoids and is involved in mtDNA transcription[23]. In the control, TFAM co-localised with DNA along the mitochondria (Fig. 2D). In the BAS-2 treated cells, there was evidence of TFAM and DNA colocalising outside of the mitochondria. The amount of TFAM/DNA cytosolic foci was quantified by manual counting of foci positive for TFAM and DNA and negative for Mito-Tracker signal. There was a dose-dependent increase in the cytosolic foci upon BAS-2 treatment (Fig. 2E). This was confirmed in the HDAC6 K/D cells, with a similar increase in the amount of TFAM/DNA cytosolic foci found in the HDAC6 K/D compared to the sgSCR control (Fig. 2F, G). Together, these data highlight that HDAC6 inhibition with BAS-2 or HDAC6 K/D causes significant changes to mitochondrial ultrastructure, including enlarged mitochondria, loss of cristae structure, and an increase in cytosolic mtDNA.

Mitochondrial cristae were markedly affected following HDAC6 disruption (Figs. 1, 2). As cristae are susceptible to damage by reactive oxygen species (ROS) produced by the electron transport chain[24,25], we next assessed mitochondrial ROS (mtROS) by MitoSOX Red staining in BT-549 cells. Indeed, mtROS was elevated post-BAS-2 treatment, as shown by live-cell confocal imaging (Fig. 3A, B). Next, we confirmed the increase in mtROS quantitatively by measuring MitoSOX Red staining by flow cytometry. There was an increase in mtROS with BAS-2 treatment in both the MDA-MB-231 and BT-549 cell lines (Fig. 3C, D). To determine if mtROS was driving cell death induced by BAS-2, we then used the mitochondrial targeted superoxide scavenger, MitoTEMPO. Mitochondrial ROS scavenging partially rescued the death induced by BAS-2-treated cells (Fig. 3E), suggesting that mtROS is a driving factor in BAS-2-induced cell death. Indeed, ROS scavenging using Mito-TEMPO rescued the disrupted cristae phenotype induced by BAS-2 in BT-549 cells (Fig. 3F). MitoTEMPO pre-treatment also effectively blocked disruptions to mitochondrial diameter (Fig. 3G) and PKMO intensity per mitochondria (Fig. 3H), which is a stain reliant on mitochondrial potential[22].

### HDAC6 interacted with mitochondrial and metabolic proteins, including FH

Based on the changes to mitochondrial ultrastructure, we returned to our HDAC6 interactome dataset to look for potential mitochondrial

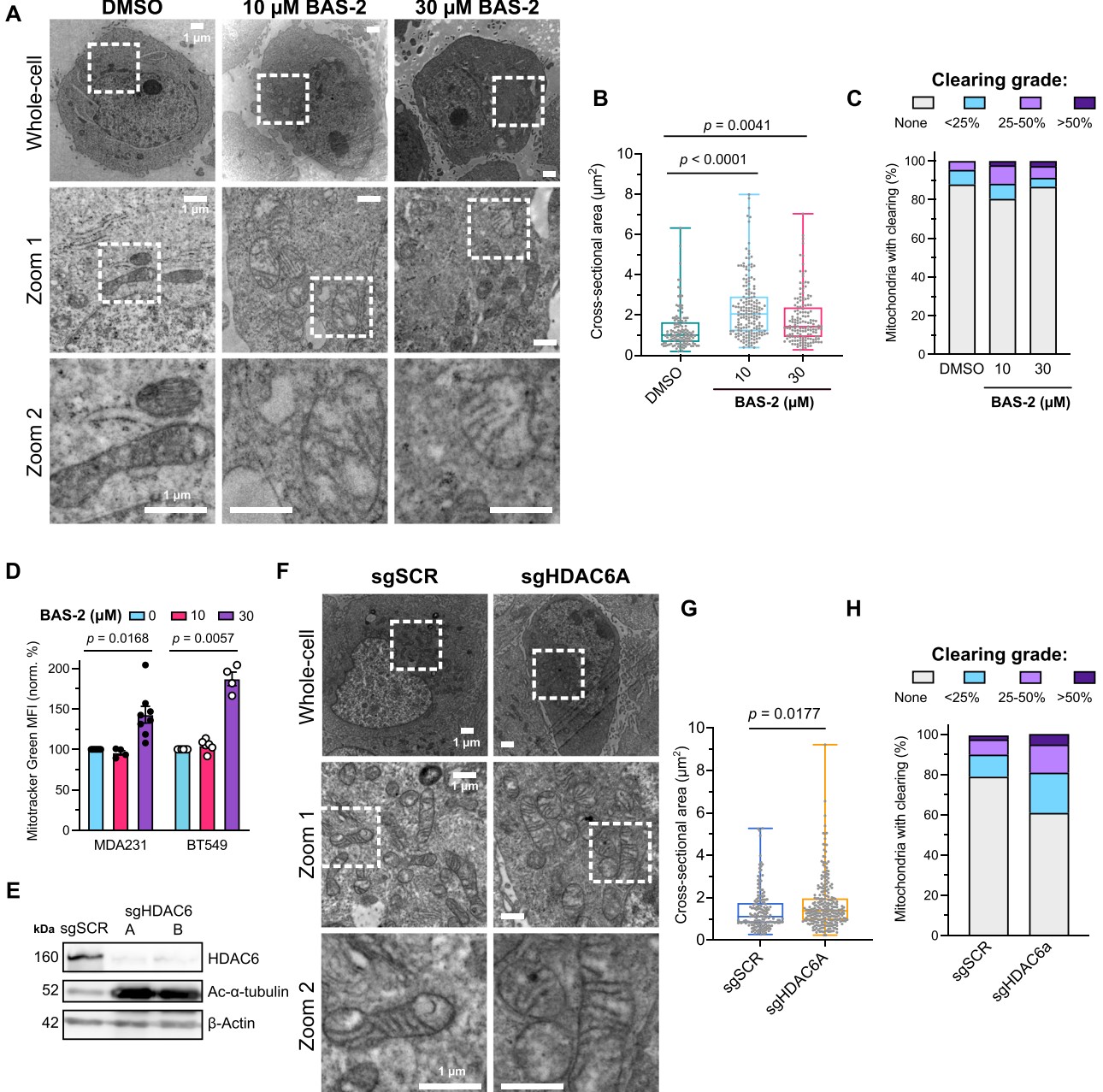

**Fig. 1 | Targeting HDAC6 activity in TNBC cells results in mitochondrial enlargement and cristae clearance. A** Representative transmission electron microscopy (TEM) images of MDA-MB-231 cells after treatment with 10 μM or 30 μM BAS-2 for 48 h. Upper panels represent whole-cell view and lower panels represent subsequent zoomed regions, as indicated by white dotted-line boxes. **B** Cross-sectional area of mitochondria manually traced from TEM images. Each grey circle represents a single mitochondrion. Data from 150-180 mitochondria from 5-8 cells per condition and one biological replicate. Significance assessed by one-way ANOVA and multiple comparisons to DMSO control. **C** Proportions of mitochondrial clearance visually graded as described in methods. **D** MitoTracker Green FM signal of MDA-MB-231 (MDA231) or BT-549 cells after treatment with BAS-2 for 24 h by flow cytometry. Data from 4-8 technical replicates from 2-4 biological repeats.

Significance assessed by one-way ANOVA and multiple comparisons against each DMSO control. Bars show mean ± SEM. **E** Validation of HDAC6 K/D mixed populations using HDAC6-targeting guides A or B using CRISPR/Cas9 in MDA-MB-231 cells by Western blot. **F** Representative TEM images of MDA-MB-231 HDAC6 K/D (sgHDAC6A) or control scramble (sgSCR) cells. **G** Cross-sectional area and **H** mitochondrial cristae clearance of MDA-MB-231 HDAC6 K/D cells. Data from 170–270 mitochondria from 10-12 cells per condition and one biological replicate. Scale bars indicate 1 μm. Box-and-whisker boxes show 25th to 75th percentiles with the median at the centre, and whiskers indicate the range of minimum to maximum values. Significance assessed by one-way ANOVA comparing to DMSO or sgSCR. Source data are provided as a Source Data file.

proteins that HDAC6 interacted with, which could lead to this altered phenotype (Fig. 4A). We generated the dataset previously by pulling down HDAC6 in MDA-MB-231 cells and measuring interacting proteins by mass spectrometry[21]. As expected, the majority of HDAC6-interactors were associated with the cytoplasm (Fig. 4B), where HDAC6 is known to predominately localise. It was less expected that

the third highest localisation group of HDAC6-interactors was the mitochondria, ranking above cytoskeleton. Interestingly, one of the most significant proteins that interacted with HDAC6 over control IgG was FH. Indeed, two recent publications have identified that FH loss induces a similar mitochondrial phenotype with evidence of cytosolic mtDNA or mtRNA[18,19]. KEGG pathway analysis of significant interactors

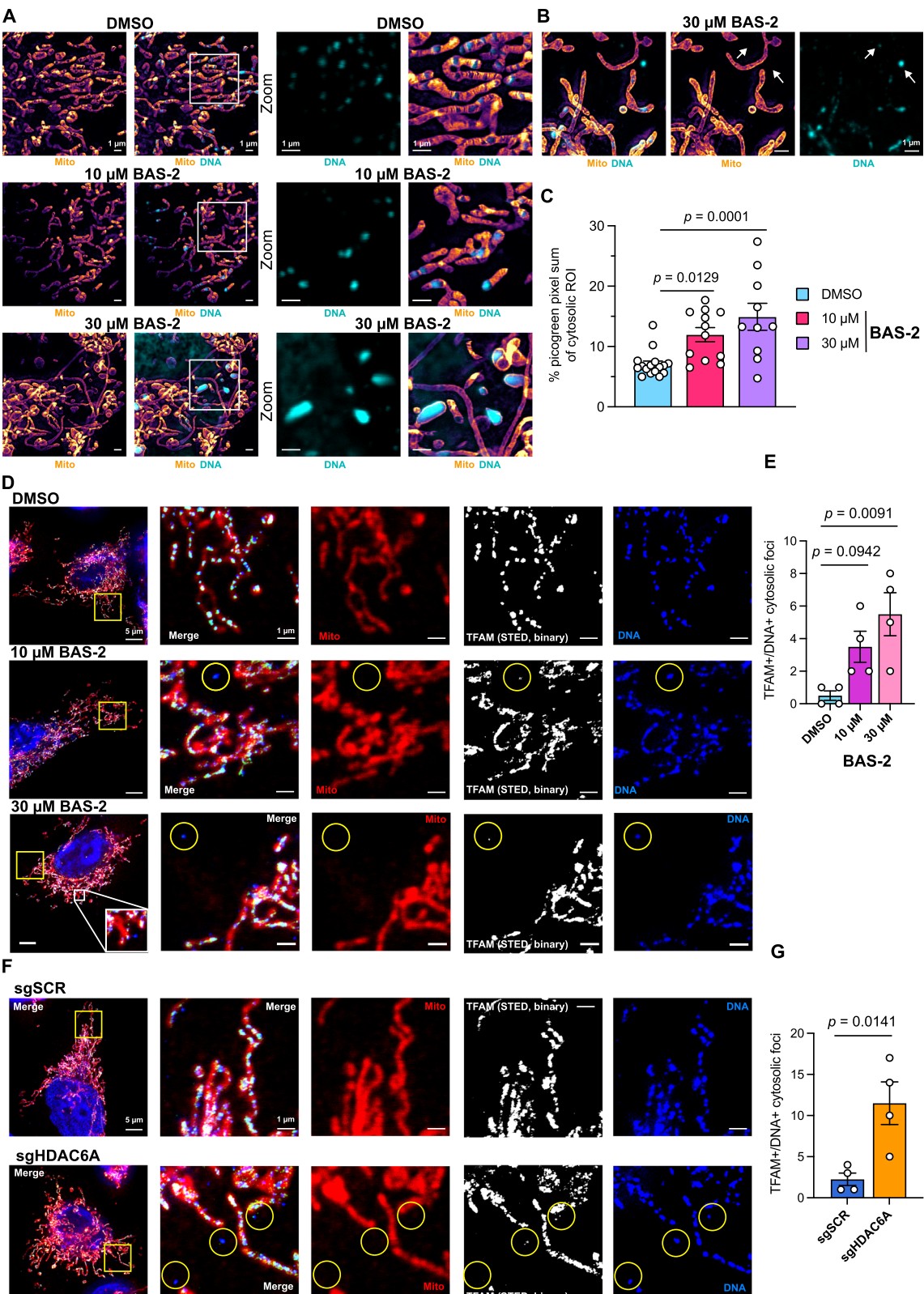

of HDAC6 (-logp > 1.3) also revealed a sizeable number of metabolic-associated proteins, ranking metabolism as the top network pathway (Fig. 4D).

To validate the interaction between HDAC6 and FH, a co-immunoprecipitation was carried out in MDA-MB-231 and BT-549 TNBC cell lines. First, an FH antibody was used for the co-immunoprecipitation and as can be seen from the Western blot,

HDAC6 immunoprecipitated, indicating direct interaction with FH in both MDA-MB-231 and BT-549 cell lines (Supplementary Fig. 2A, B). In the IgG control IP, HDAC6 was not detected. This was confirmed by the reverse pulldown of HDAC6 and assessment of FH binding (Supplementary Fig. 2C, D).

Unlike other HDAC family members, HDAC6 uniquely contains a ubiquitin-binding domain (UBD) near its N-terminus that has been

**Fig. 2 | Treatment with the HDAC6 inhibitor, BAS-2, caused live-cell mitochondrial cristae disruption by super-resolution imaging and resulted in mtDNA extrusion. A** Live-cell super-resolution images (STED) of BT-549 mitochondria using the mitochondrial membrane stain PKMito Orange (PKMO, orange) and DNA dye picogreen (cyan) after BAS-2 treatment for 24 h. Scale bars show 1 μm. **B** Representative images of high-dose (30 μM) BAS-2 treatment cytosolic DNA foci. Scale bars show 1 μm. **C** Non-nuclear picogreen pixel sum intensities as a proportion of the area quantified from live-cell super-resolution images. Points represent a single image's picogreen channel pixel sum intensity. Data show 10-16 technical replicates (images) across 3 independent experiments. Bars represent mean ± SEM. **D** Representative fixed confocal images of MDA-MB-231 cells after 24 h treatment of BAS-2 stained with MitoTracker DeepRed FM, TFAM with anti-rabbit IgG Alexa

Fluor568 (AF568), and DNA (Progen) with anti-mouse IgM Alexa Fluor488 (AF488). MitoTracker and DNA-AF488 signals were acquired using standard confocal and TFAM-AF568 was acquired using STED depletion. Merged images show TFAM-AF568 signal as green, and binary threshold panels (STED, binary) show TFAM signal in white. **E** TFAM + /DNA + /MitoTracker- foci quantified. Data from 4 technical replicates (images/regions) and one biological repeat. Bars show mean ± SEM. **F, G** As D, E with MDA-MB-231 HDAC6 K/D (sgHDAC6A) and control (sgSCR). Data from 4 technical replicates (images/regions) and one biological repeat. Bars show mean ± SEM. Significance assessed by two-sided unpaired t-test to sgSCR control or one-way ANOVA with multiple comparisons to DMSO. Scale bars show 1 μm or 5 μm (**D** and **F**, left panels). Source data are provided as a Source Data file.

linked to the regulation of misfolded protein quality control and aggresome formation[26]. Therefore, we explored the possibility of HDAC6 performing ubiquitin-mediated quality control of mitochondrial proteins. To this end, we stably transfected a FLAG-tagged HDAC6 vector missing the ubiquitin-binding zinc finger (BUZ) domain (ΔBUZ-HDAC6-FLAG, Supplementary Fig. 2E). IP experiments revealed that FH could bind both endogenous HDAC6 and ΔBUZ-HDAC6, indicating that ubiquitin-binding is not the basis for the interaction (Supplementary Fig. 2F).

Next, we assessed the global proteome of MDA-MB-231 cells and the multiple myeloma cell line JJN3 following treatment with BAS-2 for 24 hr by data-independent acquisition mass spectrometry (Fig. 4E). Similarly, to our interactome dataset, the most significantly enriched pathways were associated with metabolism (Fig. 4F), and mitochondria were highlighted as a region with significantly altered proteins (Fig. 4F, G). Interestingly, multiple mitochondrial enzymes were affected by BAS-2 treatment (Fig. 4G), including downregulation of SDH components and upregulation of the mitochondrial deacetylase SIRT5 and antioxidant protein glutathione peroxidase 1 (GPX1) (Fig. 4G, Supplementary Fig. 3B, C). Validating the metabolic alterations in MDA-MB-231 cells, in JJN3 cells the most downregulated pathway in the proteome analysis was the tricarboxylic acid metabolic pathway (Supplementary Fig. 3D, E). Taken together, these data show that HDAC6 directly interacts with the TCA enzyme FH and inhibition by BAS-2 alters metabolic pathways including the TCA cycle.

Of note, FH expression in neither JJN3 or MDA-MB-231 cell lines changed after BAS-2 treatment (Supplementary Fig. 3B–D). This was confirmed by Western blot after treatment with BAS-2 (Supplementary Fig. 3F, G). Furthermore, mRNA expression equally remained unchanged, both in the total levels and in the transcript specifically targeted to mitochondria (Supplementary Fig. 3H, I).

## HDAC6 inhibition caused an increase in fumarate

As we had observed a distinct mitochondrial phenotype, a direct interaction between HDAC6 and FH, and evidence of altered TCA cycle by global proteomics, our next aim was to determine if mitochondrial metabolism was affected by BAS-2 treatment. To assess this, we performed $^{13}C$ stable isotope tracing of glucose in MDA-MB-231 cells (Fig. 5A). Total ion counts, representing total metabolite abundance, for citrate of the TCA cycle decreased after BAS-2 treatment, indicating reduced entry of acetyl-coA (Fig. 5B). However, the following metabolites, alpha-ketoglutarate and succinate were not significantly altered. Interestingly, we saw an increase of the oncometabolite fumarate, with a reduction of the following metabolite malate, suggesting an effect of HDAC6 inhibition to the TCA cycle, potentially by blocking FH activity.

Strikingly, fractional M + 2 fumarate increased (Fig. 5G) following BAS-2 treatment, indicating the first round of the TCA cycle, while succinate (Fig. 5F) and malate (Fig. 5H) were unchanged. It was surprising, however, that both M + 2 and M + 4 fractional abundances of fumarate increased after BAS-2 treatment (Fig. 5G). This indicated that both the first and second rounds of the TCA cycle were affected. M + 3

malate also increased (Fig. 5H), indicative of decarboxylation of pyruvate to malate through pyruvate decarboxylase to replenish intermediates[27], further suggesting TCA cycle perturbation.

As it is known that fumarate can form covalent adducts on cysteine residues in a process called succination[16], we next assessed protein succination by immunofluorescence with a 2-(succino)cysteine (2-SC) antibody (Fig. 5I, J) and by Western blot (Fig. 5K). Following BAS-2 treatment for 24 hr, we observed an increase in the succination of proteins (Fig. 5I-K). This data confirms in cells that the increased fumarate, following BAS-2 treatment, is having a downstream effect causing an increase in protein succination. As a positive control, the cell-permeable derivative of fumarate, monomethyl fumarate (MMF) was used which also caused an increase in 2-SC staining (Fig. 5I). Utilising the succination signal of a dose-response of MMF treatment, the concentration of fumarate after BAS-2 treatment can be inferred (Supplementary Fig. 4A−C). BAS-2 treatment at 10 μM induced succination of proteins approximately equivalent to 25 μM MMF/fumarate (Supplementary Fig. 4C). Similarly to BAS-2, MMF also induced mtROS in both BT-549 and MDA-MB-231 cells (Supplementary Fig. 4D). Interestingly, MMF induced cell death following 24 hr treatment in the TNBC cell lines (Supplementary Fig. 4E) and enhanced the cell death induced by BAS-2 (Supplementary Fig. 4F). Combined, this data indicates that fumarate accumulation is cytotoxic to TNBC cell lines. Interestingly, upon analysis of The Cancer Genome Atlas "TCGA" data[28], FH was copy number amplified in over 40% of breast cancer patient samples and 15% of lung cancer patient samples (Supplementary Fig. 5A). This suggests that breast cancer may be particularly susceptible to perturbations in FH activity. There was also an association between survival and FH copy number amplification, with FH-amplified tumours showing a reduction in patient survival (Supplementary Fig. 5B).

Next, we aimed to confirm the reduction in FH enzymatic activity. Using a whole-cell enzymatic colorimetric assay, FH activity was significantly decreased in both the MDA-MB-231 and BT-549 cells following treatment with BAS-2 (Fig. 5L). As FH disruption induced by BAS-2 resulted in metabolic rewiring, TCA cycle alterations, and detrimental effects to mitochondria, we investigated the possibility of targeting TCA cycle compensation. M + 3 malate was increased (Fig. 5H), which is indicative of pyruvate anapleurosis via pyruvate carboxylase (PC). Therefore, we next explored inhibition of PC using ZY444 (Fig. 5M). Interestingly, ZY444 alone caused some cell death in both MDA-MB-231 and BT-549 cells (Fig. 5M, N), highlighting a vital role for PC in these cells. ZY444 exacerbated cell death induced by BAS-2 treatment (Fig. 5M), which was most significant in the BT-549 cells, demonstrating an important role for the rewiring of PC activity to support cell survival following HDAC6 inhibition.

Combined, we show that BAS-2 treatment caused alterations to TCA cycle metabolism, in particular an increase in M + 2/M + 4 fumarate. Increased intracellular fumarate was confirmed by measuring downstream succination of proteins. Metabolic rewiring as a result of BAS-2 treatment could then be harnessed using a small molecule inhibitor of PC.

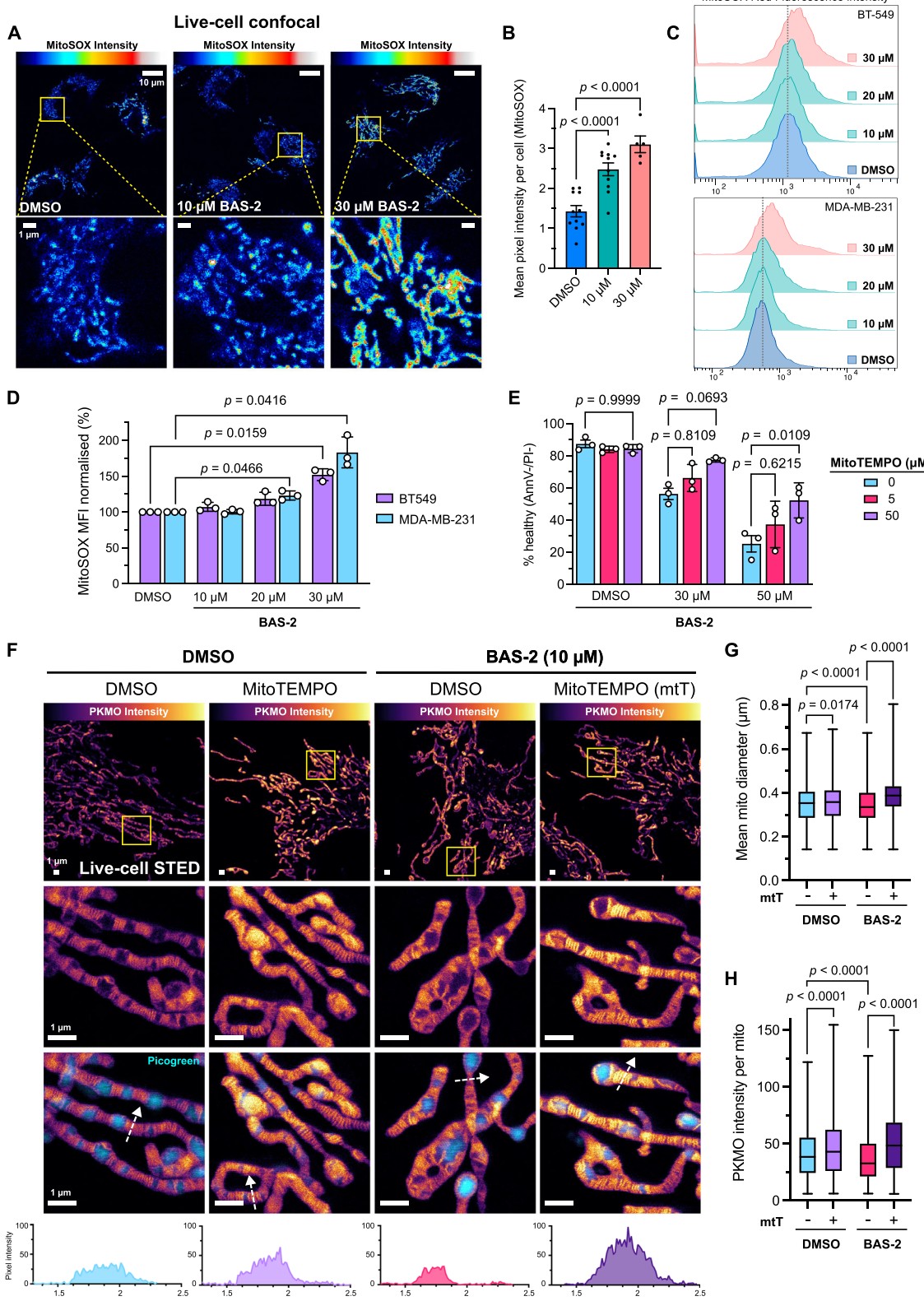

## Super-resolution 3D-STORM revealed HDAC6 interacting with FH at mitochondrial networks

While FH and HDAC6 were confirmed interactors (Fig. 4C, Supplementary Fig. 3A–D), FH can be localised in cytosolic, nuclear, or mitochondrial compartments. As such, it was still unclear where this interaction was taking place. To assess the location of the interaction, a series of standard and super-resolution imaging approaches were used. FH was first confirmed to be mainly localised in the mitochondria by confocal 3D reconstructions in MDA-MB-231 (Fig. 6A) and BT-549 cells (Supplementary Fig. 6A–D, Supplementary Movie 1). Interestingly, HDAC6 signal was associated at mitochondrial networks, as indicated by yellow surfaces (Fig. 6A, Supplementary Fig. 6B, Extended Movie 2). To validate our results from confocal imaging at higher resolution, we then utilised 3D stochastic optical reconstruction

**Fig. 3 | HDAC6 inhibition with BAS-2 increases mtROS that leads to cell death and disruption to mitochondrial cristae structure in TNBC cells.**
**A** Representative confocal images of live BT-549 cells after treatment with BAS-2 (10 μM and 30 μM) for 24 h stained with MitoSOX Red to indicate mitochondrial ROS. Lower panels show zoomed regions as indicated by yellow boxes. Scale bars show 10 μm (upper panels) and 1 μm (lower). MitoSOX Red intensity is max-adjusted and coloured using the 'Royal' colour map in ImageJ/FIJI of the scale (left to right) 0-100%. **B** Quantification of live-cell confocal images of BT-549 cells after BAS-2 treatment (24 h) where points represent mean pixel intensity of MitoSOX red channel from segmented cell regions. Data show 5-11 technical replicates (segmented cell regions) from one biological replicate. **C** Representative histograms of MitoSOX Red signal from MDA-MB-231 or BT-549 cells after 24 h BAS-2 treatment. **D** MitoSOX Red mean fluorescence intensity (MFI) of BT-549 and MDA-MB-231 cells after 24 h treatment with BAS-2 normalised to DMSO controls ($n = 3$ independent experiments). Bars represent mean ± SEM. **E** Cell death in MDA-MB-231 cells after pre-treatment with the mitochondrial ROS scavenger MitoTEMPO for 1 h and subsequent treatment with BAS-2 for 24 h, as measured by Annexin V-FITC and propidium iodide (PI) staining, where cells negative for both were designated 'healthy'. Values normalised to control and mean ± SEM graphed ($n = 3$ independent experiments). **F** Representative live-cell STED images of BT-549 cells after pre-treatment with MitoTEMPO (mtT, 50 μM), treatment with BAS-2 (10 μM) and staining with PKMO (300 nM, 30 min) for mitochondrial membrane (orange) or picogreen for DNA (cyan). Line profiles showing PKMO signal are shown below indicating the path of the white dotted arrows. Scale bars show 1 μm. PKMO intensity is max-adjusted and coloured using the 'mpl-inferno' colour map in FIJI/ ImageJ of the scale (left to right) 0-100%. **G** Box-and-whisker plots showing mitochondrial diameters from 3 confocal images per condition (images not shown) of the treatments indicated (mtT, 50 μM; BAS-2, 10 μM) from one biological replicate. Plots represent the data from 4400-5500 automatically segmented mitochondria, 1000-2000 on average per confocal image. Mitochondria were analysed as described in methods using the Mitochondria Analyzer FIJI plugin[60]. Diameters were measured on a per-mito basis. **H** Mean PKMO signal on a per-mito basis from 4400-5500 segmented mitochondria from 3 confocal images and one biological replicate. Box-and-whisker boxes show 25th to 75th percentiles with the median at the centre and whiskers indicate the range from minimum to maximum. Significance by one-way ANOVA. Source data are provided as a Source Data file.

microscopy (3D-STORM), a super-resolution technique that can routinely reach a resolution below ~30 nm. Here, we rendered images using a 20 nm pixel size.

FH was first assessed in relation to the boundaries of mitochondria using co-staining with the outer mitochondrial membrane translocase, TOM20. In BT-549 cells, the majority of FH was localised within the TOM20 boundary under basal conditions (Fig. 6B, C). Line profile analysis indicated that FH was mostly within the mitochondrial matrix at a distance of ~100 nm from the outer membrane down to distances of ~40 nm (Fig. 6C, Supplementary Fig. 7A). This roughly corresponds to the thickness of mitochondrial membranes and validates the increased resolution capabilities of STORM. The average diameter of mitochondria, as measured by the distance between TOM20 peaks, was also within the previously observed ranges (Supplementary Fig. 7A, Fig. 3F).

As FH was confirmed to be mostly contained within the mitochondria, we next assessed interactions of FH with HDAC6 by STORM. Surprisingly, points of FH-HDAC6 STORM localisations were embedded within these mitochondrial networks (Fig. 6D white triangles). To confirm this localisation, orthogonal views were created in the yz plane to observe the mitochondrial lumen (Fig. 6E). Distinct, connected circular patterns of FH can be seen, confirming that these most likely represented mitochondrial structures. Here, FH localised mostly near the edge of mitochondria. After treatment, mitochondrial networks appeared deregulated and at higher doses (30 μM) expanded (Fig. 6D, E), confirming previous results.

To measure the co-localisation of HDAC6 and FH, a recently described spatial correlation analysis was used[29] that measures distance between points and fits a Gaussian curve (Fig. 6F). This method is well-suited for super-resolution imaging. STORM localisations of FH and HDAC6 are shown in Fig. 6G, which were labelled as contributions by the analysis. When overlaid with the original image, these highlight mitochondrial networks, confirming our 3D confocal data. The total number of colocalised points, defined using a 32 nm cut-off, are shown in Fig. 6H and appear to increase with BAS-2 treatment, possibly due to disruption and condensation of mitochondrial structures. This result was validated using Spearman's analysis (Fig. 5I). A known interactor of HDAC6, HSP90, was then used to validate our analysis pipeline. Here, interactions between HDAC6 and HSP90 were shown by both Spearman's correlation and spatial cross-correlation (Supplementary Fig. 6C–F), with a more dispersed cellular localisation of HSP90.

As an intra-mitochondrial localisation of HDAC6 was unexpected, fractionation studies were then carried out. HDAC6 can be seen in the mitochondrial-enriched fraction, confirmed by high TOM20 staining in the BT-549 cells (Supplementary Fig. 7B). HDAC6 expression, as expected, was highest in the cytosolic fraction. As a proportion of FH

was also localised here, we first assessed interactions of FH and HDAC6 in the cytosol. Immunoprecipitation of FH in cytosolic fractions revealed low amounts of HDAC6 bound which did not increase after BAS-2 treatment. Next, we confirmed that HDAC6 was present in purified mitochondria (Supplementary Fig. 7D) and assessed the binding of HDAC6 and FH here. As both HDAC6[30] and FH[31] can translocate to the nucleus, we also sought to determine if an HDAC6-FH interaction was occurring in nuclear extracts. Validating our STORM analyses, HDAC6 immunoprecipitated with FH in mitochondrial fractions (Supplementary Fig. 7E). Interestingly, HDAC6 also appeared to interact with FH in nuclear extracts. To validate the nuclear interaction and assess the relative abundance of interaction in each compartment we then assessed our 3D confocal and STORM datasets. While FH did appear to interact with HDAC6 in nuclear regions by both STORM and 3D confocal imaging, the relative abundance of interaction appeared notably higher outside of the nucleus than inside (Supplementary Fig. 7G–I). To further clarify a mitochondrial interaction, we then confirmed STORM localisations of HDAC6 and FH using a smaller 2 nm pixel size (Supplementary Fig. 7J). These data suggest that the most abundant localisation of an HDAC6-FH interaction is at the mitochondria of TNBC cells.

To further assess the nature of HDAC6 with mitochondria in live cells, we next over-expressed GFP-tagged HDAC6 in MDA-MB-231 cells (MDA-HDAC6-GFP). Using PKMO staining to identify mitochondria, HDAC6-GFP can be seen associating with mitochondria (Supplementary Fig. 8; Supplementary Movie 3), though expression of the GFP vector showed significant variability (Supplementary Fig. 8A–D), appearing as highly-expressed puncta or dispersed throughout the cytoplasm.

Altogether, these data illustrate a unique regulation of FH activity in mitochondria of tumour cells by HDAC6 inhibition and that disruption leads to fumarate accumulation, protein succination, and mtROS-dependent cell death. Metabolic rewiring can then be harnessed using inhibition of PC to block the replenishment of TCA cycle metabolites. Bioinformatic interrogations point towards an amplification of FH in breast cancer that may reveal an FH dependency that can be targeted with HDAC6 inhibition.

## Discussion

Here, we report a regulation of fumarate hydratase (FH) activity in tumour cells that is mediated by the lysine deacetylase, HDAC6. Upon HDAC6 inhibition, TNBC cells exhibit dampened FH activity with simultaneous fumarate accumulation. This results in downstream protein succination, mtROS, mtDNA release, and ultimately cell death. Recent studies have linked fumarate elevation to cristae deregulation, mtROS, and mtDNA release thereby supporting these findings being fumarate-dependent[18,19]. Harnessing super-resolution imaging, we

show an interaction of HDAC6 with mitochondrial networks of FH. We also illustrate for the first time, the sub-mitochondrial localisation of FH in TNBC cells. Interestingly, 3D-STORM and spatial cross-correlation analysis suggest that mitochondrial HDAC6-FH interactions increase after HDAC6 inhibition. Rewiring of mitochondrial metabolism due to a reduction in FH activity induces a metabolic vulnerability that can be targeted using metabolic inhibitors such as the pyruvate carboxylase inhibitor, ZY444.

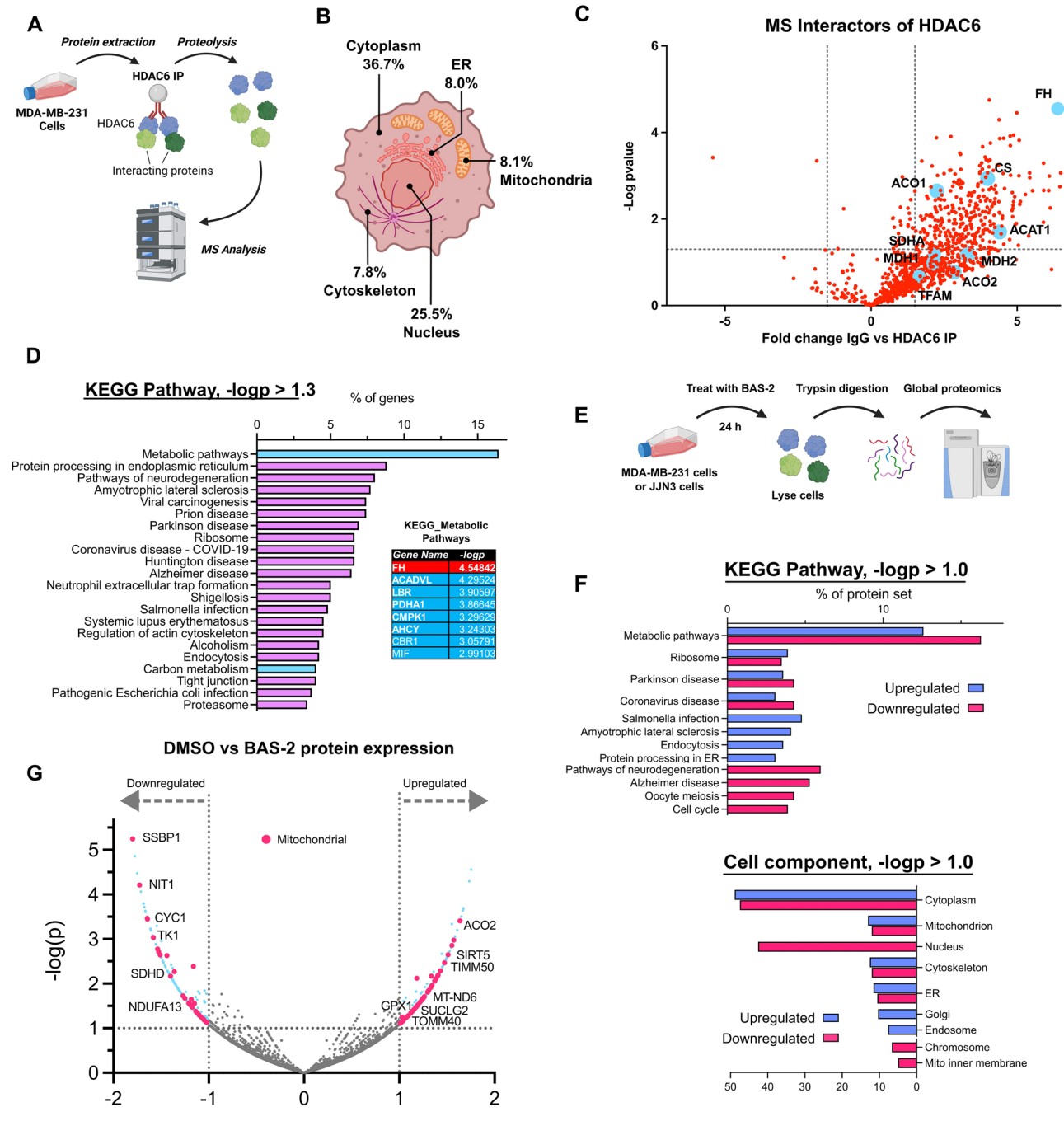

Fig. 4 | HDAC6 interacts with multiple metabolic and mitochondrial proteins, including fumarate hydratase (FH). A Schematic of mass-spectrometry-based immunoprecipitation of HDAC6 to determine interacting proteins. B Proportional localisation of interacting proteins above a probability threshold of -logp = 1.3 derived via two-tailed t-test HDAC6 IP vs IgG IP control as assessed using DAVID using the cell component descriptor. Created in BioRender[62]. C Interactors of HDAC6 compared to IgG rabbit antibody control pulldown and -logp significance value derived via t-test HDAC6 IP vs IgG IP control; a selection of metabolic- and mitochondrial-associated proteins are highlighted in blue. D KEGG pathway analysis of HDAC6-interactors above the probability threshold (t-test HDAC6 IP vs IgG IP control, -logp > 0.3) with the top interactors in the 'metabolic pathways'

subgroup shown to the right. DAVID combines KEGG pathway terms, such as different metabolic pathways, to reduce term redundancy. E Schematic of global proteomics experiment in MDA-MB-231 or JJN3 cells after 24 h treatment with 10 μM BAS-2. F KEGG pathway and cell component analysis of upregulated (blue) and downregulated (red) proteins after BAS-2 treatment in MDA-MB-231 cells. G Volcano plot of increased and decreased protein expression in MDA-MB-231 cells after BAS-2 treatment (10 μM, 24 h) by proteomics, showing log₂(fold change) vs -log(probability [p]) derived via two-tailed t-test. A -log(p) cut-off of 1 is indicated by dotted lines, and a log₂(fold change) of -1 or 1. Mitochondrial proteins are highlighted in red, and other significant proteins in blue. Source data are provided as a Source Data file.

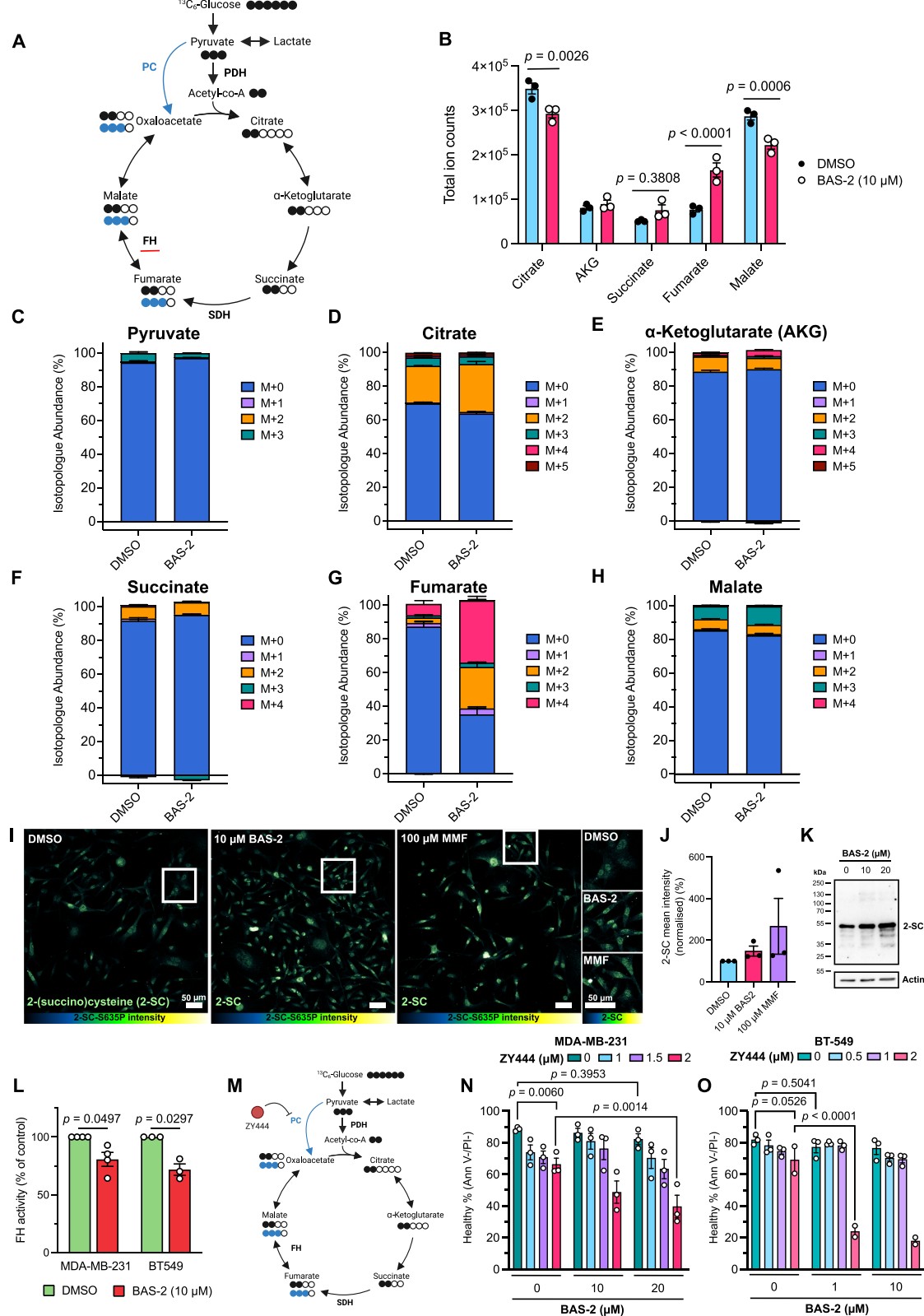

HDAC6 is a protein deacetylase[4] that can regulate enzymatic activity through deacetylation[21,32]. Indeed, increased acetylation of FH could potentially explain the reduction in FH activity, following HDAC6i. The yeast analogue, fumarase, contains a conserved lysine within its enzymatic active site[33,34]. Previously, we analysed the acetylome of TNBC cells following HDAC6 inhibition with BAS-2 or HDAC6 K/D but FH did not emerge as a target protein with increased acetylation[21]. However, in a published acetylome dataset, FH had increased acetylation following HDAC6 inhibition with tubacin[35]. Potentially, this indicates that HDAC6 may regulate the activity of FH through acetylation and further study is warranted to determine if CD1 or CD2 domain of HDAC6 is required for reduced FH activity. An alternative explanation for the increased interaction of HDAC6 and FH upon BAS-2 treatment is that the damaged mitochondria, as evidenced

**Fig. 5 | HDAC6 inhibition affects TCA cycle metabolism in MDA-MB-231 cells.**
**A** Diagram of the TCA cycle and $^{13}$C6-glucose stable isotope tracing. Associated
number of carbons in each metabolite are shown as circles where filled circles
represent $^{13}$C and clear circles endogenous $^{12}$C. Pyruvate carboxylase (PC) activity is
shown in blue. Created in BioRender[62]. **B** Total ion count values from MDA-MB-231
cells after 24 h treatment and acute, 30 m $^{13}$C6-glucose infusion. Values represent
the total abundance detected of all unlabelled and labelled fractions. Three tech-
nical replicates are shown from one independent experiment. **C–H** Relative iso-
topologue abundances of each TCA cycle metabolite including pyruvate in the
format M + x, where 'x' indicates the number of $^{13}$C carbons. **I** Representative
immunofluorescence images of MDA-MB-231 cells after 24 h treatment with BAS-2
or monomethyl fumarate (MMF) using 2-succinocysteine (2-SC) antibody with anti-
rabbit Star635P secondary to indicate protein succination. Images are max-
adjusted and coloured using the 'Green-Fire-Blue' colour map in FIJI/ImageJ with
values (left to right) 0-100%. Panels to the right show zoomed regions, as indicated
by white boxes. Scale bars show 10 μm. **J** Cells were manually segmented and mean
pixel intensity quantified. Points represent the normalised mean of three biological

repeats. 40-50 technical replicates (segmented cell regions) per biological repeat
were averaged and normalised to DMSO control. Significance by one-way ANOVA
to DMSO, which was not significant (p > 0.05). **K** Representative Western blot from
MDA-MB-231 cells after 24 h treatment with BAS-2 (10 μM and 20 μM) and immu-
noblotted for 2-SC. **L** Fumarate hydratase activity from whole cell lysates of MDA-
MB-231 or BT-549 cells after 24 h, as per manufacturer's (Sigma, MAK-206) proto-
col. Values normalised to DMSO as 100% (n = 3-4 independent experiments).
**M** Schematic of activity of ZY444 on pyruvate decarboxylase (PC). Created in
BioRender[62]. **N** Cell death by Annexin V/propidium iodide flow cytometry of MDA-
MB-231 cells after pre-treatment (24 h) with the PC inhibitor ZY444 and subsequent
BAS-2 treatment for 24 h (n = 3 independent experiments). **O** Cell death of BT-549
cells after ZY444 and BAS-2 treatment (n = 2-3 independent experiments). Two-way
ANOVA was used to assess significance in (**N, O**) with multiple comparisons com-
paring each condition against each other. Selected comparisons are shown. Points
represent independent biological replicates and the mean +/− SEM is graphed.
Source data are provided as a Source Data file.

by mtDNA in the cytosol, enable more HDAC6 trafficking inside the
mitochondria. HDAC6 was found to interact with mitochondrial
transport proteins (e.g. TOM40)[21], this may explain why FH and
HDAC6 show higher colocalisation by STORM analysis following BAS-2
treatment. Further studies are required to understand exactly how
HDAC6 enters mitochondria.

FH is a dual-localised protein, and cytosolic FH forms part of the
urea cycle by hydrolysing fumarate generated from arginosuccinate
lyase (ASL) activity[36]. Resultant malate can then re-enter mitochon-
dria through the malate-aspartate shuttle[37]. Our glucose-labelling
analyses did not show differences in glucose-derived fractions of
aspartate, although total levels did increase. Malate generated from
fumarate hydrolysis (M + 2 or M + 4 malate) did not increase either,
however there was an increase in malate from PC anapleurosis
(M + 3), where PC catalyses the carboxylation of pyruvate to
oxaloacetate[27]. Indeed, TNBC cells were sensitive to PC inhibition
alone, indicating this pathway is important for the replenishment of
TCA cycle intermediates. A similar PC-dependent anapleurotic route
has been shown to be important for the survival of lung[38] and pan-
creatic tumours[39].

HDAC6 is known to play a role in the trafficking of misfolded
proteins to aggresomes by binding to ubiquitin directly[26]. We rea-
soned, therefore, that HDAC6 may be involved in the quality control of
ubiquitin-tagged FH. However, FH was able to bind to HDAC6 lacking
the UBD. Combined, these data point towards a role of HDAC6 that
serves to support FH activity. Although the ubiquitin-binding domain
was not required for the interaction with FH, HDAC6 may still regulate
the trafficking of mitochondrial FH. Retro-translocation from the
mitochondria to the cytosol has been shown for yeast fumarase[9].
Indeed, retro-trafficking of FH towards the nucleus may be required to
maintain DNA damage repair[31], however, cytosolic HDAC6/FH inter-
actions were not evident in the STORM analysis. If HDAC6 was
important for the retro-translocation of FH, as this is a dynamic event
of potentially small stoichiometry, this may result in accumulation of
FH within the mitochondria after HDAC6 inhibition.

Fumarate hydratase deficiency is the hallmark of a genetic form of
renal carcinoma, HLRCC, characterised by heterozygous FH deletion[14].
Fumarate accumulation was shown to cause mitochondrial ultra-
structure changes, elevated ROS and a pro-tumourigenic, inflamma-
tory phenotype[8,14,17,40]. In HLRCC models, FH deletion results in
millimolar levels of intracellular and extracellular fumarate. This is
higher than what we observe after HDAC6 inhibition. However, HLRCC
is associated with a genetic deletion causing loss of FH protein,
whereas here, only a proportion of endogenously-expressed FH binds
to HDAC6. Our model has also not been genetically altered to affect FH
expression or activity. FH deletion models likely have to compensate

for fumarate accumulation to survive. Indeed, FH loss in RCC increases
mass succination of proteins such as GPX4[41], but also activates survival
factors such as NRF2 to induce antioxidant responses, including glu-
tathione (GSH) synthesis[16]. Fumarate accumulation has direct effects
through the succination of proteins but can also inhibit α-
ketoglutarate-dependent dioxygenases, including histone demethy-
lases that lead to epigenetic changes[42].

In contrast, fumarate accumulation in our tumour model
appears to be lethal, with exogenous MMF inducing mtROS and cell
death in TNBC. Interestingly, we find that FH is amplified in over 40%
of breast cancer patients and amplification is associated with poorer
overall survival in the TCGA dataset[28]. Potentially, this may indicate
that breast cancer is particularly sensitive to fumarate accumulation
and it may be a metabolic vulnerability that can be targeted.
Potential mechanisms to increase fumarate are to exogenously
increase fumarate levels by cell permeable DMF. DMF is currently
used for the treatment of multiple sclerosis and psoriasis[43]. There
are also clinical trials assessing DMF in relapsed chronic lymphocytic
leukaemia[44] and cutaneous T cell lymphoma[45]. An alternative strat-
egy to adding exogenous DMF, based on our findings, is to indirectly
increase fumarate through HDAC6 inhibition by BAS-2. Targeting
HDAC6 offers potential advantages as it can simultaneously target
glycolysis[7,21] and cancer cell motility[46].

These findings represent the first report of a direct regulation of
FH activity and homoeostasis in tumour cells. We also define HDAC6
modulation of mitochondrial metabolism and HDAC6-dependent
effects on mitochondrial ultrastructure. HDAC6 was found to inter-
act directly with FH and colocalise at intramitochondrial locations
using super-resolution STORM imaging. Inhibition of HDAC6 with BAS-
2 then leads to fumarate accumulation, downstream protein succina-
tion, and mtROS-driven cell death. HDAC6-mediated regulation of FH
activity induces a mitochondrial metabolic vulnerability that could be
harnessed for TNBC treatment.

## Limitations
A limitation of this study is the lack of assessment of glutamine flux.
Glutamine is the second most common source of carbon for cell
metabolism and a critical precursor for glutathione synthesis[1]. It would
be important to assess glutamine utilisation as we would surmise that
this is partly used to account for glycolytic and TCA cycle deficits. As
fumarate results in a reduction of glutathione (GSH) activity[40], this
pathway may directly contribute to ROS levels.

One drawback is that we have only investigated the regulation of
FH by HDAC6 in TNBC cells. It remains to be seen whether this is a
TNBC-specific interaction. Further work is warranted therefore in
other cancer subtypes, as well as in non-transformed tissue.

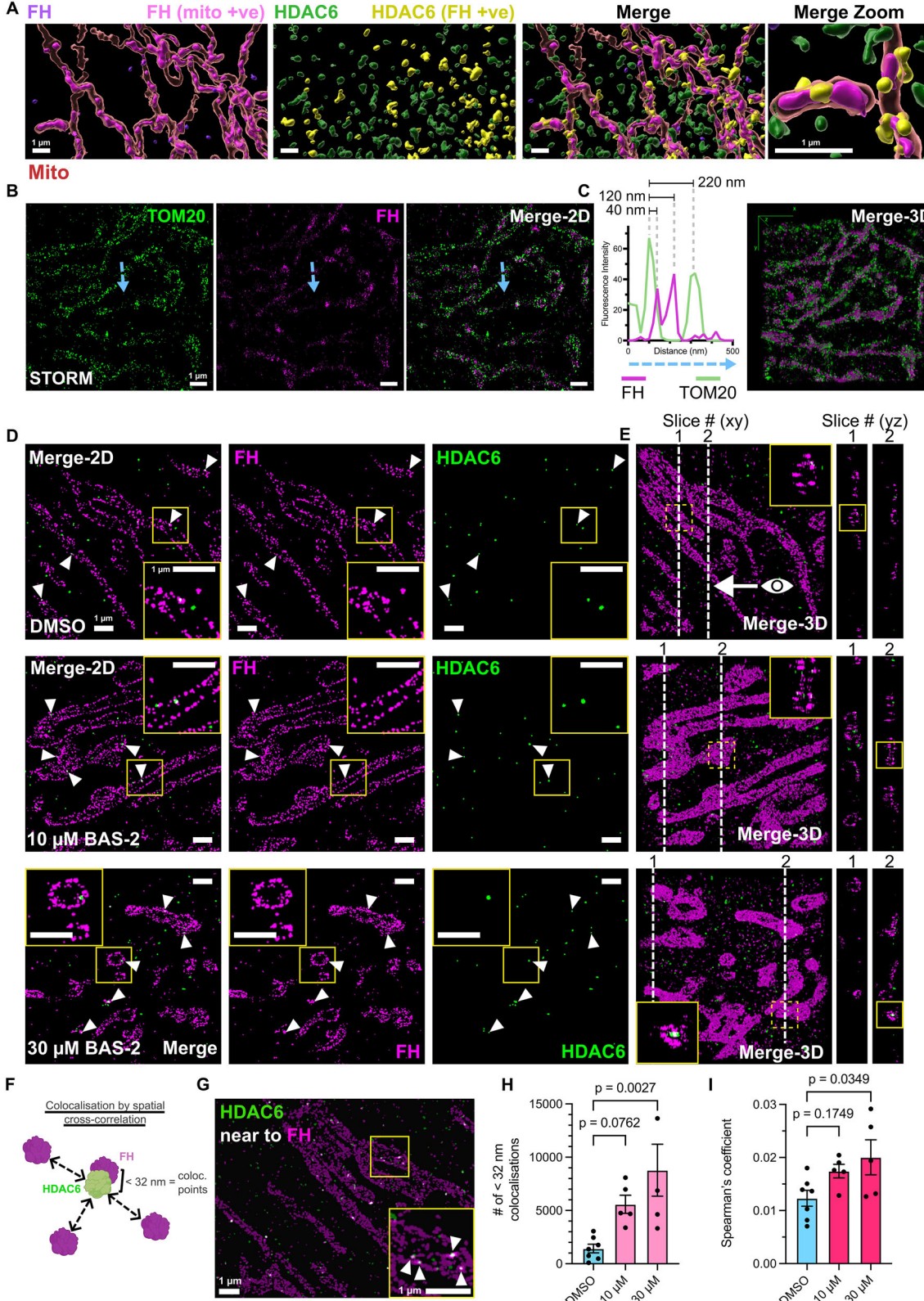

## Methods

### Cell lines & cell culture

MDA-MB-231 and BT-549 triple-negative breast cancer cell lines, JJN3, and HEK293T cells were obtained from the American Type Culture Collection (ATCC). RPMI-1640 medium (Sigma-Aldrich) was used for TNBC cells and DMEM/F12 (Sigma-Aldrich) for HEK293T. All media was supplemented with 10% foetal bovine serum (FBS), 5 mg/ml penicillin/streptomycin, and 10 mM L-glutamine. Cells were maintained at 37 °C with 5% $CO_2$ and split at 70% confluency.

Live-cell cristae investigations were performed on BT-549 cells due to a wider mitochondrial diameter and slower rate of motility. MDA-MB-231 and BT-549 cells were used to validate phenotypic alterations as occurring in TNBC cell lines. The myeloma cell line JJN3 was used to validate proteomic alterations of the HDAC6 inhibitor, BAS-2.

**Fig. 6 | FH interacts with HDAC6 at mitochondrial networks as revealed by multiple 3D imaging approaches, including super-resolution 3D-STORM.**
**A** Representative 3D reconstruction of a fixed MDA-MB-231 cell stained for mitochondria (MitoTracker DeepRed), FH, and HDAC6. Mitochondrial volume is indicated in red. FH volume overlapping mitochondrial volume (FH mito +ve) is in magenta and FH not overlapping in dark purple. HDAC6 overlapping FH (HDAC6 FH +ve) is in yellow and not overlapping in green (HDAC6 FH -ve). Scale bars show 1 μm. **B** Representative STORM images of a fixed BT-549 cell stained for FH (magenta, Alexa Fluor-647) and TOM20 (green, CF-680). Experiment was conducted two independent times with at least 2 regions imaged each time of approximately 50-70 z-slices. **C** Intensity line-profile along the blue line indicated in (**B**) and 3D reconstruction of the region in (**B**). **D** Representative STORM images of BT-549 cells after 24 h treatment with DMSO or BAS-2 (10 μM or 30 μM) showing FH (magenta, CF-680) and HDAC6 (green, Alexa Fluor647). White triangles indicate HDAC6-FH interactions at mitochondrial structures. Yellow boxes represent inlay region. Scale bars show 1 μm. Images taken from one biological repeat. **E** 3D-STORM reconstructions of regions in (**D**) with orthogonal views shown to the right. Dotted lines indicate the point at which each of the two views are taken. Yellow boxes represent inlay region and yellow dotted-line boxes indicate the region being shown. **F** Schematic illustrating the colocalisation by cross-correlation (CCC) method[29]. Created in BioRender[62]. **G** Representative contributions generated by the CCC analysis are highlighted, which illustrates all HDAC6 signal near to FH signal in this region. 5 DMSO regions were assessed by CCC from one independent experiment. Scale bars show 1 μm. **H** Total colocalisation points summed using a threshold of 32 nm. Each point represents one of 5 (DMSO), or 7 (10 μM, 30 μM) regions comprising 20-30 z-slices each and from one independent experiment. Two outliers were excluded using the 'identify outliers' test in GraphPad Prism. Significance by unpaired one-way ANOVA compared to DMSO control where points show mean ± SEM. **I** Spearman's correlation coefficients of the z-stacks used in (**H**) using the Coloc2 plugin (ImageJ/FIJI). Each point represents one of 5 (DMSO), or 7 (10 μM, 30 μM) regions comprising 20-30 z-slices each and from one independent experiment. Significance by unpaired one-way ANOVA compared to DMSO control where bars show mean ± SEM. Source data are provided as a Source Data file.

## Western blotting

Protein samples were first lysed in lysis buffer (10 mM Tris, 1 mM EDTA, 1% Triton-X100, 0.1% sodium deoxycholate, 0.1% SDS, 140 mM NaCl) containing protease and phosphatase inhibitors (Sigma-Aldrich) and quantified by BCA assay (Thermo Fisher). Samples were then denatured at 95 °C in 1X Laemmli loading buffer. 20-30 μg protein was loaded onto 10% polyacrylamide gels, and separated by electrophoresis onto 0.2 μm pore nitrocellulose membranes (2 h transfer at 90 V). Membranes were then blocked with 5% milk in tris-buffered saline with Tween20 (TBS-T) buffer for 1 h. Antibodies were incubated overnight in 5% TBS-T at 1:1,000 concentration at 4 °C. Horseradish peroxidase (HRP)-conjugated goat anti-rabbit or anti-mouse secondary antibodies were then incubated for 1 h at room temperature before being imaged via enhanced chemiluminescent substrate (Millipore) in a LAS-3000 imaging system (Fujifilm). Loading control antibodies were incubated afterwards with the same membrane. Densitometry was performed using ImageJ[47].

## Antibodies

Fumarate hydratase antibody (J-13) and HSP90 (F-8) were from Santa Cruz and 2-succino-cysteine (2-SC) from Biosynth/Cambridge Research Biochemicals (6772). Anti-DNA IgM primary antibody was from Progen (AC-30-10). Anti-DDDDK (FLAG-sequence) was from Abcam (ab1162). β-actin and HRP-secondary antibodies were from Sigma-Aldrich. Fluorophore-conjugated secondary antibodies goat anti-rabbit or anti-mouse Alexa Fluor 488, Alexa Fluor 568, Alexa Fluor 594, and Alexa Fluor 647 were from Thermo Fisher Scientific. Anti-rabbit IgG Star635P secondary was from Abberior. For STORM, donkey anti-mouse and anti-rabbit IgG secondaries conjugated to Alexa Fluor 647 or CF680 were from Jackson ImmunoResearch. All other antibodies were from Cell Signalling: HDAC6 (D2E5), acetyl-tubulin K40 (D20G3), HSP90 (#4874), TFAM (D5C8), TOM20 (D8T4N), IgG XP rabbit isotype control (DA1E), and IgG2b mouse isotype control (E7Q5L). All primary antibodies were IgG isotypes unless stated otherwise.

## Lentiviral transfection of TNBC cell lines

HEK293T cells were used to produce lentiviral particles (pMDM2.G/VSV11 and psPAX2) for CRISPR/Cas9 K/D of HDAC6 and for HDAC6-GFP expression. Mirus-LT1 (MirusBio) transfection reagent and Opti-MEM media (Thermo Fisher) were used to transfect HEK293T cells. Viral particles were then added to MDA-MB-231 or BT-549 cells using polybrene transfection reagent (8 μg/ml). Puromycin (2 μg/ml) was used to select cells for 7 days. Transfection was then confirmed by Western blot, immunofluorescence, or flow cytometry.

For HDAC6-GFP transfections, a pLVX-HDAC6-AcGFP-N1 vector was used. For CRISPR/Cas9 K/D, second-generation pLentiCRISPRv2 constructs containing scrambled single guide control (sgSCR) or HDAC6-targeting single guides (sgHDAC6) were used. All vectors were validated by Sanger sequencing and DNA gel digestion. Guides were of the following sequences: sgSCR, CCTAAGGTTAAGTCGCCCTC; sgHDAC6A, TGTGCTGAGTTCCATTACCG; sgHDAC6B, AATGGAAGAA GACCTAATCG.

## Real-time quantitative polymerase chain reaction (RT-qPCR)

RNA was extracted from $5 \times 10^6$ MDA-MB-231 cells using a Qiagen RNeasy Plus mini kit (#74134). Cells were pelleted at 500 x g, lysed in 350 μL RLT-plus reagent containing phosphatase and protease inhibitors and gDNA was eliminated using gDNA eliminator columns with centrifugation at 8000 x g. RNA was eluted in TE buffer and 1 μg used to generate cDNA using random nucleotides and reverse transcriptase (RT) enzyme (Thermo Fisher). Expression of full-length FH and FH containing a mitochondrial-targeting sequence (MTS-FH) was then determined using SYBR Green PCR Master Mix (Thermo Fisher, Applied Biosystems, #4309155) and RT-qPCR was performed in a QuantStudio Absolute Q Digital PCR system (Thermo Fisher, Applied Biosystems). Data was processed using Thermo Fisher Design & Analysis 2 software (Version 2.8.0) and presented using the -ΔΔCt method where GAPDH was housekeeping control. The following primers were used: full-length FH forward, TGC AAT AAT GAA GGC AGC AG; reverse, TGA TCC AGT CTG CCA TAC CA; MTS-FH forward, GAA ATT CTA CCC AAG CTC CCT; reverse, CGG GAG CCG AAG CTA AG.

## ΔUBD-HDAC6-FLAG Vector Transfection

pcDNA-HDAC6.ΔBUZ-FLAG (ΔUBD-HDAC6-FLAG) was a gift from Tso-Pang Yao (Addgene plasmid #30484; http://n2t.net/addgene:30484; RRID:Addgene_30484)[26]. $2.5 \times 10^6$ MDA-MB-231 cells were transfected with 15 μg plasmid DNA in lipofectamine 3000 (Thermo Fisher) for 16 h in OPTIMEM media and 10% FBS. Media was then replaced and cells incubated for 48 h. G418-sulphate/neomycin (Sigma-Aldrich) was added at 1 mg/ml with additional HEPES buffer for pH stabilisation (50 mM final) and then selected for 14 days. Media was replenished every 4 days. Western blot with FLAG antibody (Abcam) was used to validate transfection.

## Mitochondrial & cytosolic extraction

Cytosolic and mitochondrial components were separated using a mitochondrial isolation kit as per manufacturer's protocol using the reagent-based method (Thermo Fisher, Cat. #89874). Briefly, cells were lysed in a lysis buffer (reagent A) and sequentially washed with buffers B-C with centrifugation (700-12,000 x g) to separate cell components. The cytosolic component was centrifuged again at 18,000 x g before being quantified by BCA and assessed by Western blot. For nuclear and cytoplasmic extractions, an NE-PER Thermo

Scientific kit (Cat. #78833) was used. Cells were lysed in cytoplasmic extraction reagent (CER), washed, and centrifuged using a second CER or nuclear extraction reagent (NER), as per manufacturer's protocol. Buffers were supplemented with protease and phosphatase inhibitors.

## Immunoprecipitation

Cells were lysed in CHAPS immunoprecipitation buffer (0.5% CHAPS, 150 mM NaCl, 2 mM EDTA, 15 mM Tris-HCl pH 8, 50 mM DTT) with protease and phosphatase inhibitors (Sigma-Aldrich) for 30 min at 4 °C. Protein was then quantified and 600 μg protein pre-cleared with Pierce protein A agarose beads (Thermo Fisher) for 1 h for non-specific binding. FH antibody or IgG mouse control were then added (5 μg) overnight at 4 °C with replenished agarose beads. Proteins were eluted in 2 X Laemmli buffer and heated at 99 °C for 5 min before being analysed by Western blot.

## Flow cytometry

Cell death was measured by FITC-conjugated annexin V (0.25 mg/ml, BioLegend) and propidium iodide (PI, 1 mg/ml, Sigma) dual-staining flow cytometry in annexin/PI buffer (10 mM HEPES, 140 mM NaCl, 2.5 mM CaCl$_2$). Flow cytometry data was acquired using a FACSCanto II (BD Biosciences) flow cytometer and BD FACSDiva software (Version 6.1.3). Cell events were first gated as forward scatter height (FSC-H) vs FSC-area (FSC-A) and then gated as events where FSC was proportional to SSC, indicating single cells (Supplementary Fig. 9). Only events within the 'single-cell' gate were assessed and acquired up to 10,000 events. Events negative for both stains were designated as 'healthy' and normalised to a control group, as indicated.

Fluorescent stains MitoSOX Red (4 μM) or MitoTracker Green FM (50 nM) (Thermo Fisher) were incubated at 37 °C for 10 or 30 min, respectively, before being washed out with warm media and resuspended in 500 μL PBS. All flow cytometry was performed with a FACSCanto II (BD Biosciences) instrument with 488 nm and 633 nm lasers.

## Transmission electron microscopy (TEM)

$1 × 10^6$ MDA-MB-231 cells were seeded 24 h before treatment. After treatment for 48 h, cells were collected and washed in PBS. These were then fixed in 3% glutaraldehyde (Agar Scientific) in cacodylate buffer for 1 h. After 3 washes of cacodylate buffer, cells were then fixed in 1% osmium tetroxide (Agar Scientific) in cacodylate buffer for 1 h on ice. After 3 more washes, samples were dehydrated in a methanol gradient and cured in LR white under UV light overnight. Hardened resins were then sectioned using a Leica EM UC6 ultramicrotome at 60-90 nm thickness and transferred to uncoated copper mesh grids (Agar Scientific). These were mounted into a transmission electron microscope (Hitachi) and imaged under 40,000 x magnification using a side-mounted XR40 Digital Camera (Advanced Microscopy Techniques Corp., Danvers, MA, USA).

Mitochondria were manually traced and cross-sectional areas calculated using FIJI/ImageJ. Density of cristae for each mitochondrion were graded as percentage of clearing: none (cristae throughout), low (up to a quarter/25% of the mitochondrion without cristae), moderate (25-50% of the mitochondrion without cristae) or severe (more than half of the area with no cristae).

## FH activity whole-cell assay

$5 × 10^5$ MDA-MB-231 or BT-549 cells were seeded 24 h before treatment. After 24 h treatment with BAS-2 (10 μM), cells were lysed and FH activity quantified as per manufacturer's protocol (Sigma-Aldrich, MAK-206). This kit creates a colorimetric NADH product that can be used to determine the rate of FH activity over time. A fumarate enzyme mix, substrate, and developer solution were added to lysed cells in a 96-well plate in duplicate. Absorbance was then read at 450 nm every minute for 60 min. Values were determined using an NADH standard curve ran in parallel. The gradient of the absorbance line over time was then proportional to the rate of FH activity. Assays were read in a CLARIOstar plate reader (BMG Labtech).

## GC-MS $^{13}$C-glucose tracing

The metabolomic dataset used here was generated previously[21,48]. MDA-MB-231 cells were treated in 12-well plates with either DMSO or 10 μM BAS-2 for 24 h, washed in PBS and supplemented with tracing medium containing 10 mM [U]-$^{13}$C-6-glucose (Cambridge Isotopes, CLM-1396) for 30 min. Metabolites were then extracted after a 0.9% ice-cold saline wash using a methanol/water/chloroform extraction method and analysed by MS.

## Mass-spectrometry assessment of HDAC6 interactors

Similarly to the metabolomic dataset, the HDAC6 interactome dataset was generated previously[21]. HDAC6 was immunoprecipitated from MDA-MB-231 cells lysed in RIPA buffer using anti-HDAC6 antibody (Rabbit, Cell Signalling) or IgG isotype control antibody (Cell Signalling) and Pierce (Thermo Fisher) protein A agarose beads. Samples were then washed and ran through a Thermo Scientific Q-exactive mass spectrometer and Dionex Ultimate 3000 (RSLCnano) liquid chromatography system. Data from Q-exactive was processed using MaxQuant and the Andromeda search engine[49].

## Mass-spectrometry proteome analysis of MDA-MB-231 and JJN3 cells

MDA-MB-231 or the multiple myeloma cell line, JJN3, were cultured in triplicate in 6 well-plates, at a seeding density of $5 × 10^5$ cells/well. All cells were lysed in radioimmunoprecipitation assay (RIPA) lysis buffer containing complete protease inhibitor cocktail tablets (Sigma-Aldrich, USA). The supernatant was collected in a new tube, following centrifugation at high speed (~30,000 x g) for 10 min at 4 °C. A standard SP3 protocol was carried out as follows 50 μg protein was diluted in lysis buffer containing 4 M urea, 50 mM ammonium bicarbonate, and 5 mM calcium chloride. Samples were reduced in 0.2 M of dithiothreitol (DTT) for 15 min at RT. 4 mM iodoacetamide (IAA) was added and incubated for 15 min at RT in the dark. 5 μL of hydrophilic and 5 μL of hydrophobic beads were mixed and washed several times in MS-grade water. Beads were added to a Deep-Well plate in a KingFisher Duo Prime with lysates and Pierce™ Trypsin Protease, MS Grade (Thermo Fisher, 90058) at 0.5 μg/μL. In other wells, 80% ethanol was added for washing. Following overnight preparation of 8 h digestion and 4 °C storage, samples were transferred to new tubes with 0.1% formic acid, dried by speed vacuum, and stored at −20 °C until use.

Samples were analysed on a Bruker timsTof Pro mass spectrometer connected to an Evosep One liquid chromatography system[50]. All data were acquired using data independent analysis parallel accumulation serial fragmentation (dia-PASEF)[51]. Data are available via ProteomeXchange with identifier PXD057865.

## Bioinformatic analysis for MDA-MB-231 and JJN3 proteomics

Data was acquired using dia-PASEF and was analysed using DIA-NN 1.9 (Data-Independent Acquisition by Neural Networks). The *Homo sapiens* subset from the Uniprot Swissprot database (reviewed) was used to generate a spectral library within DIA-NN (library-free mode)[52].

Data was then analysed using Perseus software (version 2.0.7.0). Label-free quantification (LFQ) data was imported and transformed to log$_2$. Normal distributions of all samples were confirmed as a quality control step. Samples were then filtered based on valid values with a threshold minimum of 3 values in at least one group. Volcano plot was constructed with an FDR of 0.05. Hierarchical clustering heatmaps were constructed with Euclidean distance based on z-score normalisation of the data.

## Analysis of TCGA datasets

The cBioPortal was used to download FH gene alteration data[53–55], utilising the pan-cancer analysis of whole genomes dataset[28]. The Xena platform was used to generate survival curve data based on FH copy number amplification[56] using the same dataset[28].

## Sample preparation for fixed-cell microscopy

For fixed-cell confocal microscopy, MDA-MB-231 or BT-549 cells were seeded onto 18 mm diameter coverslips (#1.5H thickness) in 12-well plates at 65-70,000 cells/well 24 h before treatment. After treatment, cells were fixed in 4% paraformaldehyde in PBS (15 min, RT), permeabilised with Triton X-100 (0.5%, 15 min, RT), blocked with 5% bovine serum albumin (BSA) in PBS, and primary antibodies incubated overnight 4 °C in 5% BSA. Fluorophore-labelled secondaries diluted in 5% BSA/PBS were then incubated for 2 h at room temperature and, if necessary, primary/secondary steps repeated. Coverslips were mounted onto glass slides using Prolong Diamond Antifade Mountant (Thermo Fisher).

## Sample preparation for live-cell confocal and stimulated-emission depletion (STED) microscopy

For live-cell microscopy, BT-549 cells were seeded onto 8-well high-wall glass bottom chambers (#1.5H thickness, Ibidi) at 8000 cells/well 24 h before treatment. After 24 h treatment, wells were incubated with staining buffer containing PKMitoOrange (PKMO, GenvivoTech/Spyrochrome, 300 nM) and picogreen (Thermo Fisher, 0.1%) or MitoSOX Red (4 μM) for 15-30 min. Dyes were washed out with warm media and cells kept at 37 °C/5% $CO_2$ during acquisition.

## Confocal and STED image acquisition

A Leica Stellaris 8 STED 3D Falcon system equipped with a DMi8 inverted microscope and LASX software (version 4.6.0.2709) was used for all confocal and STED acquisitions. A variable white light laser (WLL, Leica) was used for excitation and a 775 nm STED laser for depletion. Three hybrid detectors (HyD S1, HyD S2, HyD X3; Leica) were used to collect signal, with HyD S2 for all STED acquisitions. The microscope was equipped with an oil-immersion 100X objective (HC PL APO CS2) with a numerical aperture (NA) of 1.4 and a 20X dry objective (0.75 NA). All confocal and STED images were acquired using the 100X objective, unless otherwise stated. Brightfield images were simultaneously acquired using the Trans-PMT mode of a non-STED channel.

## Acquisition parameters for confocal and STED imaging

Fluorophore-conjugated secondary antibodies and fluorescent stains were acquired using the following settings (excitation [nm], collection window [nm]): PKMO (590, 600-780), MitoTracker DeepRed FM (640, 650-750), Alexa Fluor-568 (577, 585-630), Alexa Fluor-594 (590, 570-640), Alexa Fluor-488 (488, 500-565), Alexa Fluor-555 (555, 555-600), Star-635P (635, 645-760), Picogreen (490, 500-590), MitoSOX Red (500, 500-750), and GFP (490, 500-570). Secondary-only controls were included for each secondary antibody to determine background fluorescence and minimise bleed-through. Fluorophores were detected in descending order according to excitation wavelength. The excitation WLL was used at lower intensities (1–10%) for the majority of confocal acquisitions but was increased to ~20% for live-cell super-resolution to counteract the loss of signal from STED depletion. Here, the STED laser was used at 80% intensity and alignment routinely performed within the LASX software. Detectors were all in analogue mode with variable gains of 1–10. Fixed-cell samples were acquired frame-by-frame and live-cell line-by-line. STED acquisition of PKMO lines were immediately followed by confocal acquisition of picogreen lines into a separate detector. A line accumulation of 3–4 was used for all STED channels and a line average of 1–4 for confocal channels.

A pinhole of 0.6-0.8 airy units (AU) and a scan speed of 200 Hz were used for all confocal and STED acquisitions. Pixel dwell times for the 100X objective were in the range (mean) of 1.6–3.5 (2.5) μs and 0.7 μs for the 20X objective. Average pixel sizes for confocal images were 70-90 nm², and 13-20 nm² when using STED. Pixel sizes and resolutions were automatically determined using the 'optimise' function within the LASX software. For 3D acquisitions, a z-step size of 0.2 μm was used for a total of 35-40 z-steps. For time-lapse imaging, a time interval of 1.026 min was used for a total of 15 frames.

## Stochastic optical reconstruction microscopy (STORM)

BT-549 cells were seeded onto 20 mm diameter coverslips in 12-well cell culture plates at 65,000 cells/well before being treated for 24 h with BAS-2 at the doses indicated. Samples were fixed and labelled like for confocal microscopy but using secondary donkey anti-mouse and anti-rabbit IgG (Jackson ImmunoResearch) that were conjugated to Alexa Fluor 647 or CF680, respectively, and post-fixed with 2% paraformaldehyde for 10 min. Samples were imaged on an inverted custom-build microscope in photoswitching buffer (0.2 M Tris pH 8.2, 10 mM NaCl, 0.5 mg mL⁻¹ glucose oxidase, 0.1 mg mL⁻¹ catalase, 4% (w/v) D(+) glucose, 35 mM mercaptoethanolamine) using 63×/1.45 NA oil immersion objective (Olympus), 640 nm excitation and 405 nm activation lasers. The emission path contained an astigmatic lens and was split (Beamsplitter T 665 LPXR, Chroma) to acquire two images in parallel on an sCMOS camera (Orca Flash 4.0, Hamamatsu) at 50 s⁻¹. Movies of 100,000-250,000 frames were processed in SMAP[57], including localisation using a spline PSF based on bead measurments[58], grouping of localisations spanning >1 frames, ratiometric assignment of channels, drift correction, filtering by localisation precision (< 30 nm) and log-likelihood ratio (LLR > -0.8), Gaussian rendering (using the localisation precision as half-width) and export of rendered 3D stacks for further analysis.

## Immunofluorescence and confocal imaging analyses

All image processing, unless otherwise indicated, was performed using the FIJI distribution package[59] of ImageJ[47] (version 2.14.0/1.54 f). For determining mitochondrial origin of cytosolic DNA, DNA-positive, TFAM-positive, and MitoTracker DeepRed FM-negative foci were manually counted and quantified as foci per image from 4 images each condition from 1 biological replicate. Colocalisation analyses were performed using either the colocalisation by cross-correlation[29] or Coloc2 plugins. In the cross-correlation analysis, distances between points are measured and a Gaussian curve fitted. Points within the distances of the fitted curve are labelled by the analysis as contributions. In the case of FH and HDAC6 signals, the most common distance and peak of the curve was beyond the resolution limit of the STORM system and therefore, total colocalising points summed instead of the Gaussian curve. 20-30 z-stack images were analysed per acquisition region for colocalisation analyses of 5-7 STORM regions. Cross-correlation analyses used distances between voxels and these were summed above the indicated threshold for each acquisition region. Coloc2 analyses used a Costes threshold regression and 10 Costes randomisations.

For assessment of cytosolic DNA signal from super-resolution STED images, picogreen-only channels were first masked to omit nuclear regions and large cytoplasmic DNA collections. Histograms were then generated showing pixel intensity distribution. A histogram was also generated per image of a cytosolic region to determine background signal and was used as the minimum intensity cut-off. A pixel intensity of 0 was used to demarcate cell boundaries. Cytosolic pixel intensities were then summed and values normalised as a percentage of the region assessed. Area was determined by the number of pixels.

Mitochondrial parameters from confocal images were determined using the 'Mitochondria Analyzer' extension, version 2.1.0[60]. A

**Table. 1 | Extensions and plugins in ImageJ/FIJI used to generate 3D visualisations**

| 3D Method | FIJI Extension | Settings & processing |
|---|---|---|
| 3D reconstructions | 3D Viewer[61] | Resampling factor = 1. Different rotations exported separately. |
| Orthogonal Slices | Stacks … Orthogonal Views | XY, XZ, and ZY views exported separately and arranged in Inkscape 1.2. |
| Maximal projections | Stacks … Z-Project | Maximal intensity projection type. |

block size of 1.05 μm was used and a c-value of 13 for threshold offset, as determined using the '2D threshold optimise' function. Descriptive statistics of mean branch diameter and mean PKMO intensity were then measured per mitochondria.

### 3D reconstruction & volume analyses

Series of z-stack images were processed for 3D visualisations. These were generated either using extensions in ImageJ/Fiji (Table 1) or using Imaris (9.8.2, Oxford Instruments).

The 'generate surfaces' function in Imaris was used to render surfaces using fixed-cell 3D z-stack acquisitions of MDA-MB-231 or BT-549 cells stained with MitoTracker DeepRed, HDAC6 (Alexa Fluor-568), and FH (Alexa Fluor-488). A grain size of 0.08 μm was used with smoothing and background subtraction (local contrast) switched on and the diameter of the largest sphere set to 0.531 μm. Surfaces were then filtered using (in order): threshold, surface area, voxel number, and mean contributing channel intensity. After generation, FH surfaces were then coloured by the statistic 'overlapped volume ratio to mitochondria surface' (-3.5–1.0, min-max) and the 'Ultra' lookup table (LUT). HDAC6 surfaces were coloured either by filter ('intensity mean of FH channel' and 'shortest distance to FH surface') or by statistic 'overlapped volume ratio to FH surface' (0.0063-1.0). Further details are listed in the Source Data file.

### Statistical analysis

All statistical analyses were performed in GraphPad Prism 9. To compare two groups, t-tests were used and one-way ANOVA for more than two. Tests were all two-tailed and paired or unpaired, as indicated. Two-way ANOVA was used to compare two or more sets of independent variables. Bars indicate mean ± standard error of the mean (SEM), unless otherwise indicated. Diagrams and schematics were generated using BioRender.com. Figures were prepared in Inkscape 1.2.

### Reporting summary

Further information on research design is available in the Nature Portfolio Reporting Summary linked to this article.

## Data availability

All data supporting the findings of this study are available within the article and its Supplementary Information files. Source data are provided with this paper as a Source Data file. The mass spectrometry proteomics data have been deposited to the ProteomeXchange Consortium via the PRIDE partner repository with the dataset identifier PXD057865. Processed mass spectrometry data is available within the Source Data file. Source data are provided with this paper.

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

## Acknowledgements

The authors would like to thank the following funding agencies for their support: Science Foundation Ireland (19/FFP/6461) support AR, DAR and CD; Irish Research Council Laureate Award (IRCLA/2022/2822) support TNC and CD'A; RCSI-Fulbright joint PhD (24/396/A1) supports MH. Confocal and STED images were acquired in the RCSI Super Resolution Imaging Consortium (SRIC) funded by Science Foundation Ireland (18/RI/5723). The Comprehensive Molecular Analytical Platform (CMAP) under The SFI Research Programme, reference 18/RI/5702, supported mass spectrometry research reported in this publication.

## Author contributions

A.R. performed experiments, analysed data, and wrote the manuscript. T.N.C. conceived the project, designed experiments, and co-wrote the manuscript. C.M.D., C.D'A., Y.W., D.A.R., C.K., K.E.R.H., M.G., B.C., K.W., M.H., and I.S. performed experiments and/or analysed data. T.L. and Z.C. generated reagents. J.H.M.P., I.S., M.M., and E.K. analysed data and contributed to manuscript editing.

## Competing interests

The authors declare no competing interests.
