## [Peer Review file · Nature Communications]

INHIBITION OF HDAC6 ALTERS FUMARATE HYDRATASE ACTIVITY AND MITOCHONDRIAL STRUCTURE

Corresponding Author: Professor Triona Ní Chonghaile

Version 0:

Reviewer comments:

Reviewer #1

(Remarks to the Author)

In the manuscript titled “A novel regulation of fumarate hydratase activity through histone deacetylase 6 binding,” Roe and colleagues report that HDAC6 inhibition or silencing in two triple-negative breast cancer cell lines causes mitochondrial defects, mtDNA release in the cytosol, and mitochondrial ROS. Mechanistically, they provide evidence that HDAC6 interacts with several TCA enzymes, particularly Fumarate Hydratase (FH). The authors demonstrate that HDAC6-FH interaction occurs at the mitochondria and that HDAC6 inhibition reduces FH activity and consequently increases fumarate and downstream protein succination.

This study proposes an exciting connection between HDAC6 and mitochondrial FH, with possible implications for tumour biology. However, some additional experiments are required to support the authors’ conclusions.

Major critiques:

1. The manuscript's central claim is that HDAC6 directly controls FH function at the mitochondria; however, how HDAC6 governs FH activity is unclear and should be determined. For instance, it is unclear whether HDAC6 interaction with FH or its deacetylase activity is required to regulate FH enzymatic activity. Indeed, the reduced FH activity may be independent of its interaction with HDAC6, and it may be due to secondary changes at the protein or mRNA levels. To rule this hypothesis out, the authors should determine whether FH mRNA and protein levels are affected by HDAC6 inhibition or silencing. In addition, a quantification of acetylation levels of immunoprecipitated FH upon HDAC6 inhibition/silencing and an evaluation of FH enzymatic activity upon acetylation should be provided.
2. Although HDAC6 is expected to be a cytosolic protein, its interactome is enriched in nuclear proteins. Since FH is also partially localised in the cytosol and mitochondria, the interaction between FH and HDAC6 may occur outside the mitochondria. This hypothesis is consistent with the observation that the levels of mitochondrial HDAC6 bound to mitochondrial FH are minimal (see following comment). The authors should determine whether FH and HDAC6 interact in other subcellular compartments, particularly the nucleus.
3. The authors suggest that HDAC6 interacts with FH in the mitochondria matrix using imaging; however, these conclusions should be validated by orthogonal approaches. For instance, the authors could demonstrate the HDAC6-FH interaction using crosslinking or Blue Native-PAGE. Also, due to the very small portion of HDAC6 interacting with FH in the IP (Figure 4F and 4G), performing an IP from mitochondrial fractions would be more appropriate.
4. 2SC levels were assessed only using immunofluorescence; however, the data shows a high level of variance, making it difficult to conclude about the 2SC increase upon BAS-2 treatment. Moreover, the selected images (Figure 5I) do not represent the described increase in 2SC levels. Additional methods of detection, such as WB, would support the conclusion that 2SC is increased upon HDAC6 inhibition.
5. The ROS part is confusing. It is unclear whether the authors propose that the increase in ROS is causally connected to the mitochondrial phenotype or is secondary. If the former, the authors should determine whether antioxidants such as mitoTempo rescue the defects in mitochondrial morphology or mtDNA release. If the latter, this part does not add much mechanistic insight to the work.

Minor critiques:

1. In general, there are some inconsistencies with the reported mitochondrial phenotype. First, In Figure 1A, BAS-2 leads to swollen mitochondria at 10 uM concentration, but the mitochondrial phenotype is very different at 30 uM (mitochondria appear smaller and condensed). Similarly, in Figure 1F the mitochondrial phenotype upon HDAC6 deletion is swollen on one panel, while in the other, the same intervention leads to highly condensed mitochondria. Along similar lines, in Figure 2, cells treated with BAS or deleted for HDAC6 do not show a swollen mitochondrial network. The same concerns apply to the mitoSOX staining in Figure 3, which shows highly elongated mitochondria upon BAS-2. These results suggest either that the mitochondrial phenotype is weak and variable or that the choice of representative images was poor.
2. The proteome experiment in Figure 4 was performed in JJN3 (myeloma cell line), while all the other experiments were done in MDA-MB-231 and BT-549 (breast cancer cell line). A rationale for this change should be explained, or proteomes should be obtained from MDA-MB-231/BT-549.
3. Extended data Figure 1E: the WB does not represent the quantification done in panel F. Indeed, the representative WB of Ac-tub seems to decrease at 30µM of BAS-2 treatment whereas the quantification shows an increase.
4. Extended data Figure 6A: the MDA-MB-231 HDAC6-GFP expressing cells show diffuse cytosolic localization opposite to the HDAC6 GFP-puncta observed in (E). This discrepancy should be clarified

Reviewer #2

(Remarks to the Author)

Reviewer #3

(Remarks to the Author)

Fumarate Hydratase (FH) is a metabolic enzyme that catalyzes the conversion of fumarate to malate in the citric acid cycle. Additionally, it is a tumor suppressor. Loss of FH expression or activity is a driver of aggressive renal cancer. Despite the role FH plays in tumorigenesis and inflammation, the mechanisms by which FH is regulated remain poorly understood. To address this gap in knowledge, the authors set out to better characterize the regulation of FH. Previous work by this group has shown that genetic loss of Histone Deacetylase 6 (HDAC6) or inhibition of HDAC6 activity led to alteration glycolysis rates. To continue this work, the authors looked at other metabolic pathways that may be altered under HDAC6 loss. The authors used transmission electron microscopy (TEM) to evaluate alterations to mitochondrial structure under HDAC6 inhibition and loss. They found that mitochondrial cross sectional area increased as well as mitochondrial size. The authors also found that HDAC6 inhibition induced mitochondrial DNA (mtDNA) release as well as increased mitochondrial reactive oxygen species (mtROS), indicating that there is mitochondrial dysfunction under loss of HDAC6 loss. Using proteomic mass spec analysis of HDAC6 pull down interactors, the authors looked for mitochondrial proteins that interacted with HDAC6. They found that FH was among the most significant hits of the mitochondrial interacting proteins. This interaction was confirmed by IP western blot. The authors then assessed TCA alterations by GC-MS. They found that there was significant increase in Fumarate and decrease in malate upon HDAC6 inhibition. Lastly, the authors examined FH-HDAC6 interaction using 3D-STORM imaging.

Overall, this paper shows that HDAC6 is a critical regulator of mitochondrial metabolic pathways by interaction with the TCA metabolite fumarate hydratase. Loss of FH activity due to HDAC6 inhibition led to accumulation of mtROS and release of mtDNA. This caused alterations in mitochondrial shape and composition. This paper offers insight into the role of HDAC6 in control of metabolic pathways. Some claims of the abstract and the paper go beyond the data provided (e.g. if you want to claim this is an interesting target in cancer metabolism, you have to show its a target by inhibition and cancer relevant effects). Some other technical concerns about specific experiments in the manuscript should be fixed. All together, the work is novel, interesting and well done and would be appropriate for publication with minor revisions.

1. Some claims are uncited/unreferenced. For example, "At concentrations of BAS-2 that can cause cell death, there is evidence of hyperfused mitochondria along with an increase in mtDNA in the cytosol" contains no reference to a figure or to the literature.
2. The authors state that HDAC6 is unusual due to its cytosolic localization and that "HDAC6 is known to predominately localize [in the cytosol]." Additionally, they show that HDAC6-FH interaction does not occur in the cytosol, rather exclusively in the mitochondria. The movement of HDAC6 to the mitochondria is unclear.
3. Figure 5C-H show the isotopologue distribution for all metabolites from m+0 to m+5. However, some metabolites (e.g. pyruvate, succinate, fumarate, etc.) do not have five carbons. This may confuse some readers; I would recommend changing to the appropriate number of carbons per metabolite.
4. Figure 5A shows SDHA alone as being responsible for the catalysis of succinate to fumarate. While SDHA is the subunit responsible for this conversion, all subunits of the SDH complex are required for activity of the complex. Recommend changing to be "SDH" instead of "SDHA".
5. Figure 5B shows the total ion counts for TCA intermediates after 13C6 Glucose infusion. It is unclear if this is the m+0

isotopologue being displayed or a sum of isotopologues.

6. Figure 5I-J data is highly variable to the point of being an unconvincing experiment. Including a decrease of signal intensity at the highest concentrations of HDAC6 inhibitor. Not sure this effectively conveys evidence of alterations to succination due to HDAC6 inhibition. An alternative experiment or a better optimized or powered experiment should be conducted.

7. FH Activity Assay methods portion lacks a reference to where the FH activity assay was purchased, only "manufacturer's protocol".

Reviewer #4

(Remarks to the Author)

In the manuscript, "A novel regulation of fumarate hydratase activity through histone deacetylase 6 binding," Roe and colleagues unravel a novel link between fumarate hydratase and HDAC6 at the site of mitochondria in triple-negative breast cancer cell lines. The group has collected a large amount of results using an impressive multi-modal approach; however, some key results remain unverified. Additional imaging is needed to fill in the gaps in the analysis and support all reported conclusions. Considering that these are cell line experiments, it is a reasonable expectation that the imaging be performed more wholly and robustly. Specifically, the quality and analysis of the 3D-STORM data should be sharpened.

Major comments and questions

For Figure 1, more TEM images are requested to confirm cisternae damage since it is not obvious. Authors should annotate evidence of cristae damage in the TEM images in order to state that; "HDAC6 inhibition ... causes significant changes to mitochondrial ultrastructure," (line 123) or "mitochondrial cristae were markedly affected following HDAC6 disruption" (line 127). Also, the authors should discuss the reason why they record ultrastructural mitochondrial expansion in response to low dose BAS-2 but do not observe this difference with MitoTracker staining (Fig. 1D)

For Figure 2, the live-cell STED images are excellent and complementary to the TEM data. In Figure 2A - B, can the whole cell view of this staining be shown, or is this not possible? Does the picogreen signal require STED or can it be captured with conventional fluorescence? In Figure 2 D and F, the cytosolic TFAM is difficult to detect. Consider showing a binary threshold image to illustrate this phenomenon. Also, please explain why STED was not also used for fixed cell imaging.

The Figure 3A MitoSOX confocal live images lack quantification and should include a less toxic (10 μ M) BAS-2 treatment condition; both are needed to support the flow cytometry data.

Lines 132-133 describes a BAS-2 dose-dependent mtROS effect in BT-549 cells, and line 135 states that "mitochondrial ROS scavenging could rescue BAS-2 treated cells,"; both are overstatements based on Figure 3C. The effect occurs in these cells is only when using 30 μ M of inhibitor, which is toxic to most cells based on Figure 3D.

The HDAC6-FH interaction was discovered by reanalyzing an old dataset, but the reverse pulldown verification in Extended Data Figure 2 is a rigorous addition. While the graphic in Figure 4B is useful, in both Figure 4B and D, the KEGG results are difficult to read. Why do so many genes need to be listed in the figure?

STORM was used to pinpoint sites of HDAC6-FH interactions at mitochondria. In Extended Data Fig. 4a, the whole cell HDAC6 signal is not detectable. Please add new images for readers to assess the cellular localization of HDAC6. Even if the resolution is poor, this is essential data. On a related note, although the "D2E5" is a good HDAC6 antibody, the authors get surprisingly weak signal with their immunofluorescence. Can this be explained?

Line 247 states that "FH was likely within the mitochondrial matrix at a distance > 40 nm from the other membrane," in reference to the line profile in Figure 6B. For clarity, "FH" should be written as "FH STORM molecules" or "FH STORM localizations". However, the whole statement should be reconsidered since the line profile is a snapshot of the FH localization inside a single mitochondrion. The measurements - even estimated differences - are not meaningful unless repeated and aggregated.

In Figure 6D, again possibly due to the staining, it is difficult to tell that these HDAC6 images are actually STORM reconstructions. Were the HDAC6 localizations all so densely clustered? Line 254 stating that "points of FH-HDAC6 interactions were identified," should be described more accurately as "instances of closely localized FH and HDAC6 STORM molecules" or something similar. Line 259 states that "after treatment with BAS-2, FH appeared more associated with mitochondria and HDAC6," in reference to Figure 2E. Please clarify the meaning here. It is not evident that FH is localized anywhere else in the cell but the mitochondria based on the fluorescence images.

The data in Extended Data Figure 6 are not additive and seem to be incomplete possibly due to technical issues. Line 277 states that "HDAC6-GFP can be seen associating with mitochondria," yet there is only 1 single visible HDAC6-GFP punctum shown in this whole dataset. The HDAC6-GFP approach is potentially a strong addition. Is live imaging essential? Why not use it in a fixed cell analysis with a GFP enhancer reagent and super-resolution to help support the Fig. 6 findings?

Finally, the immunogold images should be removed from the manuscript since there are no immunogold labeling controls provided.

The Methods section is too brief. Manufacturer's protocols should not be cited. For example, the "Mitochondrial and cytoplasmic extraction" and "FH activity whole-cell assay" subsections should include more useful details. There is no live-cell confocal microscopy methodology. Please report all STORM acquisition and analysis parameters as well as details about how the STORM data were visualized, sorted and refined.

Minor comments and questions

Line 31-32 states that "HDAC6 inhibition or knockdown resulted in mitochondrial cristae damage" followed by line 74, "HDAC6 can alter mitochondrial structure," which is confusing. The authors should clarify the cause of damage.

The Introduction is written in an unusual manner. It provides too little information to establish the biological premise of the study. Instead, it drives toward a review-style summary of the latest findings and observations. Additional premise could be added to the Introduction section to boost interest.

Throughout the manuscript, the cell lines used are difficult to track. The authors should consider describing the cell types, their rationale for using them for different types of experiments and report any measured differences between cell lines.

The top image for Fig 1A for 30 um BAS-2 is blurry (potentially unfocused). Is there a better example image available?

Fig 2C, E, G seem to contain too few points in these data sets. With cell lines and cell staining protocols, it seems reasonable that these analyses should be filled with more data points.

Fig2D - which DNA stain was used?

Fig 2E and G - what criteria were used for selecting TFAM/DNA cytosolic foci?

Line 128 - "[mito] cristae are prone to damage by ROS" references citation 22. Can the authors clarify here? The cited paper doesn't discuss cristae damage specifically. Is there literature evidence that ROS specifically causes structural damage to mitochondria cristae?

The authors should consider switching the order of Extended Data Figures 4 and 5. It is confusing to go from 3 to 5 and then go back to 4 later on.

Lines 249-251 is a description that mitochondria diameters were "within the previously observed ranges." This is vague and should be clarified.

Fig 4I - consider labeling some of the up- and downregulated processes in the graph.

Fig 6D - could the images be magnified even more to see the single molecule localization?

Fig 6G - the images are difficult to see.

Fig 6H - why were data from the imaging fields aggregated? How were these data normalized? Please provide more details.

Fig 6H, I - consider comparing the localization of HDAC6 to a control target (like microtubules) to enhance the rigor of the colocalization data.

Line 281 references Extended Data Fig. 5F-G but it should be Extended Data Fig. 6.

The Discussion unfortunately did not provide an explanation or hypothesis for why the HDAC6-FH interactions may increase after chemical HDAC6 inhibition.

Line 415 - Figure 3, legend - is missing the label "(D)".

Lines 506-508 contains unnecessary bolded text.

Extended Data 4 - authors should consider using the same color arrows in this Figure. The use of different colored arrows gives the impression that readers are supposed to be seeing different targets.

Lines 598 and 653 - the buffer name is spelled differently. The Line 653 spelling is correct.

Line 711 - "St. et al.," is confusing; potentially a citation? Please double check.

Line 771 - The "R" should be capitalized in "Jackson Immunoresearch."

Line 969 - Please double check this last citation.

The authors should provide uncropped western blot scans for review if they have not done so already.

Reviewer #5

(Remarks to the Author)

Version 1:

Reviewer comments:

Reviewer #1

(Remarks to the Author)

The authors addressed most concerns satisfactorily, and the work has improved. Well done for the significant effort! As a minor note, since the authors could not confirm the mechanism of regulation of FH by HDAC6 and most of the functional work has been conducted using an HDAC6 inhibitor, the title may not reflect the overall conclusions of the paper. In other words, the current work convincingly shows that HDAC6 inhibition causes the described changes in mitochondrial ultrastructure and metabolism, but it is less clear whether, under basal conditions, HDAC6 regulates or is required to govern these parameters. The title should therefore better reflect the results.

Reviewer #2

(Remarks to the Author)

Reviewer #3

(Remarks to the Author)

The authors adequately responded to all major comments.

Reviewer #4

(Remarks to the Author)

All comments have been adequately addressed, and the authors have prepared an excellent revised manuscript.

Reviewer #5

(Remarks to the Author)

Response to Reviewers

Dear Reviewers,

We would like to thank the reviewers for taking the time to assess our work and for their constructive suggestions. The reviewers stated that the paper showed “*an exciting connection between HDAC6 and mitochondrial FH*” “*all together this work is novel, interesting and well done*” and “*impressive multi-modal approach*” was carried out. In addressing the points raised by the reviewers’, we feel we have significantly strengthened the manuscript. The new data strongly supports an interaction of FH with HDAC6 within the mitochondria of TNBC cells and provide further support of HDAC6-inhibitor induction of fumarate-mediated succination. Below is a point-by-point response to the reviewers’ comments; the reviewers’ comments are in italics and the response is in regular type.

Reviewer #1

Major critiques:

1. The manuscript's central claim is that HDAC6 directly controls FH function at the mitochondria; however, how HDAC6 governs FH activity is unclear and should be determined. For instance, it is unclear whether HDAC6 interaction with FH or its deacetylase activity is required to regulate FH enzymatic activity. Indeed, the reduced FH activity may be independent of its interaction with HDAC6, and it may be due to secondary changes at the protein or mRNA levels. To rule this hypothesis out, the authors should determine whether FH mRNA and protein levels are affected by HDAC6 inhibition or silencing. In addition, a quantification of acetylation levels of immunoprecipitated FH upon HDAC6 inhibition/silencing and an evaluation of FH enzymatic activity upon acetylation should be provided.

We fully agree with the Reviewer that the perturbations to FH activity may arise from altered protein or mRNA expression of FH. As such we have conducted mass-spectrometry proteomics, Western blotting, and q RT-PCR to show that FH levels remained unchanged in all settings (Figure 4 and S. Figure 3). Further, when assessing mRNA we determined total FH mRNA, along with the mitochondrially-targeted sequence (MTS) transcript. Neither changed following BAS-2 treatment and it showed that the majority of FH mRNA is of the MTS-type, supporting our immunofluorescence and fractionation data indicating that the majority of FH is within the mitochondria. Western blot and qRT-PCR data have been included in S. Figure 3F-I.

Unfortunately, despite our best efforts to assess the acetylation status of FH, the heavy band of IgG frequently obscured results of the immunoprecipitation. Below are shown both FH immunoprecipitation and acetyl-lysine immunoprecipitation using mouse or rabbit primaries for both AcK and FH. Figure C shows a potential dose-dependent increase in acetylation of FH using the AcK mouse antibody, however it overlapped with IgG and it was not reproducible with the AcK rabbit antibody in D.

However, FH has been reported as acetylated at multiple residues (in Choudhary et al. 2009 and Scholtz et al. 2015). Further, Scholtz indicates that FH acetylation increases upon Tubacin (HDAC6 inhibitor) treatment. In addition, our MDA-MB-231 proteomic data indicate an upregulation of SIRT5, a mitochondrial deacetylase (Fig 4G). Therefore, despite not directly showing FH acetylation, this appears a likely reason for the results observed and is stated in Lines 370-377.

Figure 1: Measuring FH acetylation through immunoprecipitation. (A) Immunoprecipitation (IP) using mouse lysine acetylated antibody following treatment with DMSO, 10 μM and 20 μM BAS-2 for 24 hr. The Western blot was then probed for FH expression. (B) Is a repeat of the IP in (A) (C) IP with FH antibody following treatment with BAS-2 for 24 hr and the Western blot was probed with mouse acetylated lysine antibody. (D) same as (C) but a rabbit acetylated lysine antibody was used. (E) IP with the rabbit acetylated lysine antibody and the Western blot was probed with FH. (F) Attempted using acetylated lysine beads to IP interacting proteins and probed for FH.

2. Although HDAC6 is expected to be a cytosolic protein, its interactome is enriched in nuclear proteins. Since FH is also partially localised in the cytosol and mitochondria, the interaction between FH and HDAC6 may occur outside the mitochondria. This hypothesis is consistent with the observation that the levels of mitochondrial HDAC6 bound to mitochondrial FH are

minimal (see following comment). The authors should determine whether FH and HDAC6 interact in other subcellular compartments, particularly the nucleus.

We agree with the reviewer's point that both proteins may be interacting in non-mitochondrial compartments and have therefore performed additional cell fractionations to address this. As can be seen in S. Figure 7B-E, HDAC6 was found localised in mitochondrial fractions. Furthermore, FH and HDAC6 interacted in purified mitochondria, which was confirmed using SDHA/TOM20 markers for mitochondrial fractions. We then assessed whether HDAC6 and FH interacted in the nuclear compartment. Interestingly, we did find an interaction of HDAC6 and FH in the nuclear fraction, in addition to FH binding to PARP (S. Figure 7D-E). However, it is important to highlight that the nuclear fractions were not completely clean, with some evidence of potential cytosolic and mitochondrial contamination.

We next assessed our confocal and STORM datasets to determine the proportionality of this interaction. We have now included a 3D reconstruction of an MDA-MB-231 cell showing a top-down view of a nuclear region and overlapping HDAC6 and FH signal, in addition to the FH signal coloured differently when not overlapping mitochondria (S. Figure 7F-G). FH-interacting HDAC6, shown in yellow, appeared far more abundant outside of the nucleus, whereas FH not interacting with mitochondria is coloured in darker purple and can be seen within the nucleus. Supporting the IP data, there was a proportion of colocalisation of FH with HDAC6 in the nucleus by STORM (S. Figure 7H).

Next, we assessed the cytosolic compartment and while we found an interaction of FH and HDAC6, this interaction was comparatively weak as compared to IgG control (S. Figure 7C), in contrast to mitochondrial IP (S. Figure 7E). This may therefore be secondary to HDAC6's role as an aggresome trafficker of ubiquitin-tagged misfolded proteins during translation. We argue that whilst an interaction within the nucleus is interesting, a comparably higher proportion of interaction appears to occur in the mitochondria. This, coupled with mitochondrial-specific defects, mtROS, mtDNA and altered ETC complex expression, would support that a mitochondrial interaction is of more significance. Nevertheless, this is an important point that we have highlighted in Lines 327-335.

3. The authors suggest that HDAC6 interacts with FH in the mitochondria matrix using imaging; however, these conclusions should be validated by orthogonal approaches. For instance, the authors could demonstrate the HDAC6-FH interaction using crosslinking or Blue Native-PAGE. Also, due to the very small portion of HDAC6 interacting with FH in the IP (Figure 4F and 4G), performing an IP from mitochondrial fractions would be more appropriate.

We thank the reviewer for this comment and hope that the mitochondrial fractionation and immunoprecipitation above suffice for an orthogonal approach. A sizeable amount of HDAC6 interacted with purified mitochondrial FH (S. Figure 7D-E).

4. 2SC levels were assessed only using immunofluorescence; however, the data shows a high level of variance, making it difficult to conclude about the 2SC increase upon BAS-2 treatment. Moreover, the selected images (Figure 5I) do not represent the described increase in 2SC levels. Additional methods of detection, such as WB, would support the conclusion that 2SC is increased upon HDAC6 inhibition.

We thank the reviewer for these comments and agree that the variability and lack of supporting approaches were detrimental to our conclusions. Elevated succination was

therefore confirmed using WB extracts (Figure 5I). We have also repeated immunofluorescence investigations with a reduced magnification objective (20X vs 100X in the original Figure), as well as further repeats of high-magnification 100X images for analyses. The original 100X images in Figure 5 and quantification have now been replaced with images showing lower magnification and more cells, which hopefully better illustrates a global increase in succination, with some notable 'hyper-succination' cells.

5. The ROS part is confusing. It is unclear whether the authors propose that the increase in ROS is causally connected to the mitochondrial phenotype or is secondary. If the former, the authors should determine whether antioxidants such as mitoTempo rescue the defects in mitochondrial morphology or mtDNA release. If the latter, this part does not add much mechanistic insight to the work.

We agree that this section could have been clearer and have revised this figure with additional experiments (Figure 3). mtROS scavenging could indeed rescue the mitochondrial cristae phenotype by live-cell STED imaging and is therefore most likely to be preceding cristae alterations induced by BAS-2 treatment. PKMO intensity was elevated by MitoTEMPO alone, indicating increased mitochondrial potential (PKMO is potential dependent). From this, it appears that MitoTEMPO preserves mitochondrial membrane potential and ultrastructure and blocks damage induced by BAS-2. We have previously shown that BAS-2 induced a loss of membrane potential (Dowling et al. 2019), which is likely the point of protection by MitoTEMPO. We have now revised the section and added Lines 159-162 in light of this data.

Minor critiques:

1. In general, there are some inconsistencies with the reported mitochondrial phenotype. First, in Figure 1A, BAS-2 leads to swollen mitochondria at 10 uM concentration, but the mitochondrial phenotype is very different at 30 uM (mitochondria appear smaller and condensed). Similarly, in Figure 1F the mitochondrial phenotype upon HDAC6 deletion is swollen on one panel, while in the other, the same intervention leads to highly condensed mitochondria. Along similar lines, in Figure 2, cells treated with BAS or deleted for HDAC6 do not show a swollen mitochondrial network. The same concerns apply to the mitoSOX staining in Figure 3, which shows highly elongated mitochondria upon BAS-2. These results suggest either that the mitochondrial phenotype is weak and variable or that the choice of representative images was poor.

We must apologise, as the reviewer is correct that our choice of representative images was poor. At times we were demonstrating variance rather than the most representative images for the phenotype. To address, this point we have gone back through the images and chosen the most representative image.

2. The proteome experiment in Figure 4 was performed in JN3 (myeloma cell line), while all the other experiments were done in MDA-MB-231 and BT-549 (breast cancer cell line). A rationale for this change should be explained, or proteomes should be obtained from MDA-MB-231/BT-549.

We thank the reviewer for this comment on an alternative cell line and agree that a supporting TNBC cell line would be appropriate to include. Therefore, we have now performed MS proteomics on the MDA-MB-231 cells after BAS-2 treatment, which supports our data in that multiple metabolic pathways are changed, including that of the TCA cycle and

mitochondrial metabolism (Figure 4F-G). We do, however, still believe that the inclusion of an alternative, non-TNBC cell line, is an important result regarding the effects of HDAC6 inhibition in different cell types and whether metabolic effects are TNBC cell-specific. Therefore, we have retained this data but now as S. Figure 3D-E and also included Lines 199 and 202 to reflect this point.

3. Extended data Figure 1E: the WB does not represent the quantification done in panel F. Indeed, the representative WB of Ac-tub seems to decrease at 30 μ M of BAS-2 treatment whereas the quantification shows an increase.

Similar to the above, we agree that the Western blots here were not the best choices and most representative. We have now included different blots that better represent the quantification of data. S. Figure 1B-C, E-F.

4. Extended data Figure 6A: the MDA-MB-231 HDAC6-GFP expressing cells show diffuse cytosolic localization opposite to the HDAC6 GFP-puncta observed in (E). This discrepancy should be clarified

We thank the reviewer for this comment and apologise for the thresholding discrepancy in this image. The reason for the differential expression of HDAC6-GFP is simply that there was heterogeneity in cells following overexpression of the GFP vector (seen in flow data of S. Figure 8B, D). We have added lines 343-345, and 715-716 to reflect this, and also updated images in what is now S. Figure 8.

Reviewer #2:

Reviewer #3:

1. Some claims are uncited/unreferenced. For example, "At concentrations of BAS-2 that can cause cell death, there is evidence of hyperfused mitochondria along with an increase in mtDNA in the cytosol" contains no reference to a figure or to the literature.

We apologise for this oversight and have now called out Figure 2A (for hyperfused mitochondria) and Figure S1A and D (for 30 μ M BAS-2 causing death) in Lines 133-135 showing this data.

2. The authors state that HDAC6 is unusual due to its cytosolic localization and that "HDAC6 is known to predominately localize [in the cytosol]." Additionally, they show that HDAC6-FH interaction does not occur in the cytosol, but rather exclusively in the mitochondria. The movement of HDAC6 to the mitochondria is unclear.

The majority of this point overlaps with point 1 by Reviewer 1 so please see above. Regarding the movement of HDAC6 to the mitochondria, we argue that the kinetics of this through the mitochondrial membranes, matrix trafficking, folding etc. are beyond the scope of the current study. Unfortunately, there is also currently very little in the published literature regarding the dynamics of FH movement into mitochondria and nothing on HDAC6.

3. *Figure 5C-H shows the isotopologue distribution for all metabolites from m+0 to m+5. However, some metabolites (e.g. pyruvate, succinate, fumarate, etc.) do not have five carbons. This may confuse some readers; I would recommend changing to the appropriate number of carbons per metabolite.*

We would like to thank the reviewer for this point and agree that this may confuse some readers. We have now adjusted the shown isotopologue abundances to reflect this in Figure 5C-H.

4. *Figure 5A shows SDHA alone as being responsible for the catalysis of succinate to fumarate. While SDHA is the subunit responsible for this conversion, all subunits of the SDH complex are required for the activity of the complex. Recommend changing to be "SDH" instead of "SDHA".*

We agree with this point and have changed the figure to simply show SDH.

5. *Figure 5B shows the total ion counts for TCA intermediates after ¹³C6 Glucose infusion. It is unclear if this is the m+0 isotopologue being displayed or a sum of isotopologues.*

We apologise for this wording. The total ion counts represent the raw abundances detected by the MS quantification, which would include all labelled and unlabelled fractions (M+0, M+1, etc.). We have added the statement ‘...representing total metabolite abundance...’ to Line 216 and also clarified this in the legend of Figure 5 to make this point clear.

6. *Figure 5I-J data is highly variable to the point of being an unconvincing experiment. Including a decrease in signal intensity at the highest concentrations of HDAC6 inhibitor. Not sure this effectively conveys evidence of alterations to succination due to HDAC6 inhibition. An alternative experiment or a better optimized or powered experiment should be conducted.*

Similar to a point raised by Reviewer 1 – we have now shown succination by WB, lower magnification images of 2-SC, as well as additional quantifications, updated as Figure 5I-K. All of these data now show global increases in succination signals, which support our conclusions.

7. *The FH Activity Assay methods portion lacks a reference to where the FH activity assay was purchased, only "manufacturer's protocol".*

We apologise for this oversight and have added Sigma-Aldrich, MAK-206 to Line 554 and Line 863.

Reviewer #4:

Major comments:

1. *For Figure 1, more TEM images are requested to confirm cisternae damage since it is not obvious. Authors should annotate evidence of cristae damage in the TEM images in order to state that; "HDAC6 inhibition ... causes significant changes to mitochondrial ultrastructure," (line 123) or "mitochondrial cristae were markedly affected following HDAC6 disruption" (line 127). Also, the authors should discuss the reason why they record ultrastructural*

mitochondrial expansion in response to low dose BAS-2 but do not observe this difference with MitoTracker staining (Fig. 1D)

We would like to thank the reviewer for their comment. We have updated Figure 1 so that the most representative TEM images are now shown as well as the whole cell region. Previously, we chose images to demonstrate the variance that is routinely found in mitochondrial imaging. In addition, in the mitotracker imaging in Figure 2D, it is clear that the TFAM and DNA staining are increased with low-dose BAS-2 treatment. Mitotracker staining was only imaged using standard confocal resolution, which is not high enough for the visualisation of cristae. However, the PKMO staining using super-resolution nicely shows that the cristae are not as crisp compared to the control and there are sections of expanded mtDNA that could potentially correlate with regions of cleared cristae in the TEM imaging.

*2. For Figure 2, the live-cell STED images are excellent and complementary to the TEM data. In Figure 2A - B, can the **whole cell view of** this staining be shown, or is this not possible? Does the picogreen signal require STED or can it be captured with conventional fluorescence? In Figure 2 D and F, the cytosolic TFAM is difficult to detect. Consider showing a binary threshold image to illustrate this phenomenon. Also, please explain why STED was not also used for fixed cell imaging.*

We thank the reviewer for these comments in relation to the modes of image acquisition and we have tried to make it clearer exactly how the images were acquired, including more substantial details on image acquisition in the methods. In Figures 2D-F, the TFAM channel was read in STED mode as the fluorophore used (AlexaFluor-568) is depletable by our STED laser. Lines 478-480 are now edited to make this clearer and we have now updated this figure with a binary threshold of the TFAM channel in Figure 2 that should better illustrate the TFAM signal present at cytosolic foci.

To the point of a whole-cell view, unfortunately, due to the high toxicity of live-cell STED imaging (from the additional STED laser and elevated intensity of initial excitation laser), further imaging of cells using STED after initial acquisition would have been confounded by photodamage, though we agree that these would have been desirable. In Figure 2, the picogreen channel was read without STED using conventional confocal (the emission of picogreen is not depletable by our STED laser) and we have edited lines 127-128 and 832 to clarify this.

3. The Figure 3A MitoSOX confocal live images lack quantification and should include a less toxic (10 μ M) BAS-2 treatment condition; both are needed to support the flow cytometry data.

We have now updated Figure 3A with a lower dose (10 μ M) that shows a similar pattern to the flow cytometry data. We have also included quantification of these images using the MitoSOX red pixel mean of each cell, which shows an increase in signal (Figure 3B).

4. Lines 132-133 describe a BAS-2 dose-dependent mtROS effect in BT-549 cells, and line 135 states that "mitochondrial ROS scavenging could rescue BAS-2 treated cells,"; both are overstatements based on Figure 3C. The effect occurs in these cells only when using 30 μ M of inhibitor, which is toxic to most cells based on Figure 3D.

We apologise for the wording of these sections and agree that it was not entirely clear the significance of mitochondrial ROS, particularly in the BT-549 cell line. We have removed the term 'dose-dependent' in Line 156 to more accurately describe a general increase in mtROS by imaging and flow cytometry.

In addition, and in response to a similar comment above, we have revised this figure with further experiments. In BT-549 cells, MitoTEMPO could preserve mitochondrial defects induced by BAS-2 (Fig. 3F-H), indicating that ROS are indeed a driving cause of this damage. We have also highlighted the cell line in Line 162.

5. The HDAC6-FH interaction was discovered by reanalyzing an old dataset, but the reverse pulldown verification in Extended Data Figure 2 is a rigorous addition. While the graphic in Figure 4B is useful, in both Figure 4B and D, the KEGG results are difficult to read. Why do so many genes need to be listed in the figure?

We apologise for the inclusion of this many proteins in Figure 4 making it difficult to read. We were just trying to include the entirety of the data but we realise now that it meant it was not as interpretable. We have now reduced the number of proteins listed, so that it should now be easier to read.

*6. In **Extended Data Fig. 4a**, the whole cell HDAC6 signal is not detectable. Please add new images for readers to assess the cellular localization of HDAC6. Even if the resolution is poor, this is essential data. On a related note, although the "D2E5" is a good HDAC6 antibody, the authors get surprisingly weak signal with their immunofluorescence. Can this be explained?*

We thank the reviewer for this comment and have updated what is now S. Figure 6A accordingly to show a single plane. In the original figure, thresholds were very strict for the HDAC6 channel to help locate significant points of interaction. We have changed this figure to show that HDAC6 is dispersed throughout the cell with the majority in the cytosol. This antibody is strong and selective for HDAC6 as validated through K/D and this was a threshold discrepancy of the image. Additional 3D visualisations have also now been provided which should help illustrate the localisation patterns (S. Figure 6B and S. Figure 7F).

7. Line 247 states that "FH was likely within the mitochondrial matrix at a distance > 40 nm from the other membrane," in reference to the line profile in Figure 6B. For clarity, "FH" should be written as "FH STORM molecules" or "FH STORM localizations".

We would like to thank the reviewer for highlighting this point. To more accurately describe the figures we have now labelled these as FH-STORM localisations in Line 287.

8. However, the whole statement should be reconsidered since the line profile is a snapshot of the FH localization inside a single mitochondrion. The measurements - even estimated differences - are not meaningful unless repeated and aggregated.

We agree with this comment and ~~we~~ have now repeated and aggregated these measurements. The new data has been updated in SFigure 7A and references in Line 290-292.

9. In Figure 6D, again possibly due to the staining, it is difficult to tell that these HDAC6

images are actually STORM reconstructions. Were the HDAC6 localizations all so densely clustered? Line 254 stating that “points of FH-HDAC6 interactions were identified,” should be described more accurately as “instances of closely localized FH and HDAC6 STORM molecules” or something similar.

The previously shown images/3D stacks had been rendered with a pixel size of 20 nm, which we have clarified in Lines 284-285. Although, we agree that this rendering may not effectively demonstrate the localisation of STORM events. For this, we have re-rendered the raw STORM data with a smaller pixel size of 2 nm and higher magnification to help demonstrate HDAC6 localisation with FH at mitochondrial structures (S.Figure 7K), and have referenced to this data in Lines 337-338.

For this response, we have also included histogram binning of HDAC6 events for this region below. However, we prefer to present rendered images in the manuscript for better visibility and to avoid confusion among readers who are not experts in super-resolution techniques.

Rebuttal Figure 1. STORM reconstructions for FH (magenta) and HDAC-6 (green) immunostainings. Shown are the localisations within a 200 nm z-slice. HDAC-6 in magnified insets (right) is shown using a histogram representation (binned localisations by pixel of 2 nm size in xy) to emphasize individual localisations. The remaining images/channels are rendered using a Gaussian representation with a Gaussian half-width equalling the localisation precision.

Line 259 states that “after treatment with BAS-2, FH appeared more associated with mitochondria and HDAC6,” in reference to Figure 2E. Please clarify the meaning here. It is not evident that FH is localized anywhere else in the cell but in the mitochondria based on the fluorescence images.

We apologise for this wording and have removed this line. We believe the same sentiment of HDAC6 and FH localisation increasing upon treatment is better supported by the colocalisation analyses at the end of Figure 6.

The data in Extended Data Figure 6 are not additive and seem to be incomplete possibly due to technical issues. Line 277 states that “HDAC6-GFP can be seen associating with mitochondria,” yet there is only 1 single visible HDAC6-GFP punctum shown in this whole dataset. The HDAC6-GFP approach is potentially a strong addition. Is live imaging essential?

Why not use it in a fixed cell analysis with a GFP enhancer reagent and super-resolution to help support the Fig. 6 findings? Finally, the immunogold images should be removed from the manuscript since there are no immunogold labeling controls provided.

We thank the reviewer for these comments. Regarding the HDAC6-GFP we have now included a re-threshold of the images and a more zoomed-out region to better illustrate HDAC6-GFP

with mitochondria (now S. Figure 8E). One caveat of the GFP-vector was the high and variable expression patterns of the transfected HDAC6 which we have clarified in Lines 345-347. As such, we have instead supported the data with more visualisation of endogenous HDAC6. The immunogold data has also been removed.

The Methods section is too brief. Manufacturer's protocols should not be cited. For example, the "Mitochondrial and cytoplasmic extraction" and "FH activity whole-cell assay" subsections should include more useful details. There is no live-cell confocal microscopy methodology. Please report all STORM acquisition and analysis parameters as well as details about how the STORM data were visualized, sorted and refined.

We thank the reviewer for this comment and agree that more descriptive information should be available for these methods. As such, we have now included more details of mitochondrial isolation and subcellular fractionation (Lines 666-676), FH activity enzymatic assay (Lines 718), and STORM methods (Lines 846). Additional details on live- and fixed-cell confocal microscopy are also provided in Lines 821.

Minor comments and questions

1. *Line 31-32 states that "HDAC6 inhibition or knockdown resulted in mitochondrial cristae damage" followed by line 74, "HDAC6 can alter mitochondrial structure," which is confusing. The authors should clarify the cause of damage.*

We have updated what is now Line 33 and Line 90 to match. Further, we have now performed additional investigations on mtROS and scavenging with MitoTEMPO in a revised Figure 3 that hopefully clarifies the source of damage.

2. *The Introduction is written in an unusual manner. It provides too little information to establish the biological premise of the study. Instead, it drives toward a review-style summary of the latest findings and observations. Additional premise could be added to the Introduction section to boost interest.*

We have reworded multiple sections in the introduction to hopefully more clearly introduce the topic and rationale. Please see the tracked changes document for the alterations made to the introduction (Lines 48-52; 55-56; 65-69; 83-88).

3. *Throughout the manuscript, the cell lines used are difficult to track. The authors should consider describing the cell types, their rationale for using them for different types of experiments and report any measured differences between cell lines.*

We agree that the reasoning for using different cell lines could have been clearer and have now added clarification for the use of MDA-MB-231 cells in Line 106 and BT-549 in Line 124-126. We have also added a statement in the Methods (Line 611) clarifying the reasoning of each cell line. BT-549 cells were used to validate our findings from MDA-MB-231 cells. The use of the JN3 multiple myeloma cell line in proteomics was to validate findings outside of TNBC, but, similar to a point by Reviewer 1, we have now performed proteomics on MDA-MB-231 cells and include this instead in Figure 4F-G, with JN3 data now in S. Figure 3D-E. In relation to cell line choice, BT-549 possess mitochondria that are more amenable to live-cell super-resolution imaging (wider diameter, slower moving, and closer to the coverslip/culture surface). We have added in Line 124-126 to clarify this.

4. *The top image for Fig 1A for 30 um BAS-2 is blurry (potentially unfocused). Is there a better example image available?*

We have now updated all TEM images with more representative examples. See updated Figure 1.

5. *Fig 2C, E, and G seem to contain too few points in these data sets. With cell lines and cell staining protocols, it seems reasonable that these analyses should be filled with more data points.*

Though we can understand the reviewer's point, we do not believe that adding more data points substantially changes either the conclusion or the interpretations of the data.

6. *Fig2D - which DNA stain was used?*

We have included in Line 477 that this experiment used a primary antibody against DNA (Progen Anti-DNA IgM, AC-30-10) that was suitable for fixed-cell immunofluorescence, which is also listed in methods under antibodies in line 630.

7. *Fig 2E and G - what criteria were used for selecting TFAM/DNA cytosolic foci?*

This was simply a manual task of counting DNA foci outside the nucleus and assessing the presence of DNA/TFAM/MitoTracker signal present. TFAM/DNA foci were TFAM and DNA positive but MitoTracker negative. This has hopefully now been clarified in Lines 141-142 and better illustrated in re-rendered TFAM in Figure 2, which shows the thresholded binary signal.

8. *Line 128 - "[mito] cristae are prone to damage by ROS" references citation 22. Can the authors clarify here? The cited paper doesn't discuss cristae damage specifically. Is there literature evidence that ROS specifically causes structural damage to mitochondria cristae?*
We apologise for this oversight in referencing and have now included an additional citation that specifically demonstrates ROS-mediated lipid peroxidation of mitochondrial cristae membranes (Line 152).

9. *The authors should consider switching the order of Extended Data Figures 4 and 5. It is confusing to go from 3 to 5 and then go back to 4 later on.*

We agree with the reviewer and have re-ordered our Extended Data Figures.

10. *Lines 249-251 is a description that mitochondria diameters were "within the previously observed ranges." This is vague and should be clarified.*

We apologise for this reference as it related to data removed from the manuscript showing mitochondrial diameter. We have now re-included equivalent measurements in Figure 3F and referenced as such in Line 295.

11. *Fig 4I - consider labeling some of the up- and downregulated processes in the graph.*

We have now moved this figure to S. Figure 3D and replaced panels in Figure 4 with MDA-MB-231 where downregulated and upregulated are now highlighted.

12. *Fig 6D - could the images be magnified even more to see the single molecule localization?*

We have re-rendered a section of these images using a pixel size of 2 nm in S. Figure 7K, in addition to a histogram binning in response to a comment above.

Fig 6G - the images are difficult to see.

We thank the reviewer for this comment. We have now increased the size and brightness of this panel, in addition to removing the contributions alone component and are now only showing HDAC6 near to FH ('contributions') as highlighted, with an additional zoomed inset region. Hopefully, these changes aid the visibility and function of this figure panel.

Fig 6H - why were data from the imaging fields aggregated? How were these data normalized? Please provide more details.

Details on this are included in lines 586-588 and further details have been included in Lines 876-880. As data were from a single biological replicate, they have not been additionally normalised beyond the automatic threshold by each analysis algorithm.

Fig 6H, I - consider comparing the localization of HDAC6 to a control target (like microtubules) to enhance the rigor of the colocalization data.

We thank the reviewer for this comment and agree that a control target would help validate our STORM analysis approach. To address this point, we have now performed STORM colocalisation of HDAC6 with a known target, HSP90, and included this in S Figure 6C-F and referenced it in Lines 316-320. As is evident from the new STORM imaging and analysis, the values of HDAC6 and HSP90 with both cross-correlation and Spearman's are within similar ranges as that observed with FH and HDAC6, validating the localisation and interaction of HDAC6 and FH.

12. Line 281 references Extended Data Fig. 5F-G but it should be Extended Data Fig. 6.

The immunogold labelling has now been removed so this point is no longer relevant.

13. The Discussion unfortunately did not provide an explanation or hypothesis for why the HDAC6-FH interactions may increase after chemical HDAC6 inhibition.

To this point, we have now included some discussion around hypothesis for why HDAC6-FH interactions may increase Lines 381-387.

14. Line 415 - Figure 3, legend - is missing the label "(D)".

We would like to thank the reviewer for identifying this.

15. Lines 506-508 contains unnecessary bolded text.

This has been corrected see Line 609 on new tracked document. We would like to thank the reviewer for identifying the unnecessary bolding of text.

16. Extended Data 4 - authors should consider using the same color arrows in this Figure. The use of different colored arrows gives the impression that readers are supposed to be seeing different targets.

We would like to thank the reviewer for the suggestion and have avoided using different colour markers throughout. These panels have now been superseded by additional 3D visualisations and a single plane image in S. Figure 6.

17. Lines 598 and 653 - the buffer name is spelled differently. The Line 653 spelling is correct.
This has been corrected on line 684 (Laemmli) of the new document.

18. Line 711 - "St. et al.," is confusing; potentially a citation? Please double check.
This should have been Sigma-Aldrich and is now corrected

Line 771 - The "R" should be capitalized in "Jackson Immunoresearch."
This has now been corrected.

19. Line 969 - Please double check this last citation.
This citation is for a conference proceeding with a url link. We have now added in the url link.

20. The authors should provide uncropped western blot scans for review if they have not done so already.
The uncropped Western blots have all been submitted in a separate Excel document.

In summary, we have addressed the majority of the points raised by the reviewers to the best of our ability. Again, we would like to thank the Reviewers and the Editor for taking the time to read our manuscript. We believe the new data has greatly strengthened our novel findings and we hope that the new version will be warmly received.